# The prognostic and diagnostic value of intraleukocytic malaria pigment in patients with severe falciparum malaria

Ketsanee Srinamon[1,6], James A. Watson [2,3,4,6] ✉, Kamolrat Silamut[1,6], Benjamas Intharabut[1], Nguyen Hoan Phu[2], Pham Thi Diep[2], Kirsten E. Lyke[5], Caterina Fanello[1,3], Lorenz von Seidlein [1,3], Kesinee Chotivanich[1], Arjen M. Dondorp [1,3], Nicholas P. J. Day[1,3] & Nicholas J. White [1,3] ✉

Severe falciparum malaria is a major cause of death in tropical countries, particularly in African children. Rapid and accurate diagnosis and prognostic assessment are critical to clinical management. In 6027 prospectively studied patients diagnosed with severe malaria we assess the prognostic value of peripheral blood film counts of malaria pigment containing polymorpho-nuclear leukocytes (PMNs) and monocytes. We combine these results with previously published data and show, in an individual patient data meta-analysis ($n = 32,035$), that the proportion of pigment containing PMNs is predictive of in-hospital mortality. In African children the proportion of pigment containing PMNs helps distinguish severe malaria from other life-threatening febrile illnesses, and it adds to the prognostic assessment from simple bedside examination, and to the conventional malaria parasite count. Microscopy assessment of pigment containing PMNs is simple and rapid, and should be performed in all patients hospitalised with suspected severe malaria.

Severe malaria is a major cause of preventable childhood death in tropical countries, particularly in sub-Saharan Africa. The continuing large death toll justifies the substantial global investment in malaria control, although relatively little attention has been paid to severe malaria in recent years[1]. Many factors affecting the prognosis of falciparum malaria have been identified in the past from detailed prospective clinical studies[2–6]. In routine clinical practice in malaria-endemic areas the initial assessment of disease severity is made in the clinic or at the bedside from the level of consciousness, vital signs, respiratory pattern and degree of anaemia. If the patient is admitted to the hospital then additional laboratory information of prognostic value may become available, notably acid-base status from plasma bicarbonate, lactate or blood gas, plasma creatinine or blood urea, and

blood glucose measurements[7]. Rapid tests for blood glucose measurement are widely available and point-of-care blood lactate measurement is increasingly available in intensive care units.

The diagnosis of malaria is made either from microscopy of stained blood slides or, increasingly, from rapid diagnostic blood tests (RDT). Unfortunately RDTs do not quantify parasitaemia and they also do not provide other information of prognostic value. However, even small, poorly equipped hospitals usually have a microscope, although the support for, and thus the quality of microscopy vary substantially[1]. Microscopy malaria parasite counts are still often semi-quantitative in reports (although this is no longer recommended), and other potentially valuable information is not assessed or reported. In assessing a patient suspected of having severe malaria, microscopy provides

[1]Mahidol Oxford Tropical Medicine Research Unit, Faculty of Tropical Medicine, Mahidol University, Bangkok 10400, Thailand. [2]Oxford University Clinical Research Unit, Hospital for Tropical Diseases, Ho Chi Minh City, Vietnam. [3]Centre for Tropical Medicine and Global Health, Nuffield Department of Medicine, University of Oxford, New Richards Building, Old Road Campus, Roosevelt Drive, Oxford OX3 7LG, UK. [4]WorldWide Antimalarial Resistance Network, Oxford, UK. [5]Center for Vaccine Development and Global Health, University of Maryland School of Medicine, Baltimore, MD, USA. [6]These authors contributed equally: Ketsanee Srinamon, James A. Watson and Kamolrat Silamut. ✉e-mail: jwatowatson@gmail.com; nickw@tropmedres.ac

important diagnostic and prognostic information obtained from counting the malaria parasites, assessing their stage of development[8], and counting the proportions of leukocytes that have ingested malaria pigment[9–11]. In many locations only a thick blood film is taken, whereas for counting high-density parasitaemias, staging parasite development and assessing intraleukocytic pigment, the thin blood film is more rapidly obtained, easier to read, and more accurate. Haemozoin, or malaria pigment, is released at schizont rupture and some of this material is phagocytosed immediately by circulating phagocytic cells. As a result the amount of this particulate material in the circulation reflects the extent of earlier schizogony. Both monocytes and polymorphonuclear leukocytes (PMNs) ingest malaria pigment. Monocytes have slower turnover in the circulation than PMNs so in blood films (or intradermal smears) malaria pigment-containing monocytes persist for longer[12]. Although there is general agreement that the proportions of PMNs or monocytes containing intraleukocytic malaria pigment are higher in more severe disease, there is disagreement about its prognostic value[9,11,13,14], notably whether this rapid microscopy assessment adds to simple clinical variables. A prospective study in 26,296 African children with a suspected diagnosis of severe falciparum malaria from the SMAC network reported that, after adjustment for the main indicators of severity (coma, acidosis, lactate and weight for age z-score), quantitation of intraleukocytic pigment-containing cell counts did not provide additional prognostic value[13]. As a result of this continuing uncertainty, intraleukocytic pigment-containing cell counts are generally not performed in routine practice, despite their speed and simplicity.

In this study we present new data on the prognostic and diagnostic value of intraleukocytic pigment in severe malaria from three large randomised clinical trials carried out or coordinated by the Oxford University Clinical Research Unit in Ho Chi Minh City, Vietnam and the Mahidol Oxford Research Unit based in Bangkok, Thailand. These are the largest prospective randomised controlled trials of Asian adults and children with severe falciparum malaria (AQ Vietnam and SEAQUAMAT)[15,16], and the largest randomised trial in African children (AQUAMAT) hospitalised with severe falciparum malaria[17]. Following a systematic review of the literature, we pooled individual patient data from over 32,000 patients clinically diagnosed with severe falciparum malaria and we assessed the prognostic value for in-hospital mortality of the proportion of pigment-containing PMNs and monocytes. This analysis of data from the majority of all patients with severe falciparum malaria studied prospectively in recent years provides conclusive evidence that assessment of intraleukocytic pigment in PMNs has important prognostic and diagnostic value in addition to that provided by a simple bedside assessment.

## Results

### Prognostic value of quantitative microscopy in three randomised trials

Quantitative pigment-containing PMN counts were available for 483 patients from the AQ Vietnam trial; 1333 patients from the SEAQUAMAT trial; and 4211 patients from the AQUAMAT trial. For the pigment-containing monocyte counts the corresponding numbers were 301, 1332, and 4186, respectively, and for parasite counts were 560, 1457, and 4786, respectively (Supplementary Fig. S1). Mortality in the patients with available and readable blood slides (where readable means that accurate counting was possible) and those without slides were comparable in the SEAQUAMAT and AQUAMAT trials. In both trials, higher parasitaemia correlated with improved slide readability (Supplementary Fig. S2). Overall 70% of slides (1010 out of 1420) were readable (i.e. countable) in patients with parasite densities less than 10,000 per $\mu$L versus 89% of slides (4283 out of 4823) in patients with parasite densities above 10,000 per $\mu$L.

We characterised the prognostic value of pigment-containing PMNs, pigment-containing monocytes, and standard malaria parasite counts in Asian adults and children and African children by fitting flexible generalised additive logistic models to in-hospital mortality (Fig. 1). The prognostic value of peripheral blood malaria parasite counts differed substantially for Asian adults and children versus African children. Low parasite counts (<1000 per $\mu$L) were associated with slightly higher than average mortality in African children (AQUAMAT, $p = 0.005$ for the spline fit). This is most likely explained by mis-classification of severe disease, i.e. misdiagnosing severe (usually bacterial) illness and incidental parasitaemia as severe falciparum malaria[18]. It has been estimated from clinical studies that approximately one third of children in sub-Saharan Africa with an initial diagnosis of severe malaria are mis-classified[19,20]. In Asian adults and children, in whom the diagnosis of severe malaria is more specific, mortality rose steeply for parasite counts > 10,000 per $\mu$L ($p = 10^{-11}$ for the spline fit) and there was no apparent increase in mortality at low parasite densities.

The prognostic value of pigment-containing neutrophils (PMNs) was consistent across Asian adults and children, and African children. Higher pigment-containing PMN counts were associated with higher mortality ($p = 10^{-8}$ and $p = 10^{-16}$ for the spline fits, respectively). The effect was greatest in the Asian adults and children, in whom the mortality in patients with 20% or more of pigment-containing PMNs was at least three times higher than in those with no pigment-containing PMNs (>30% vs <10%, respectively). By contrast pigment containing monocytes had a lower prognostic value, although the quantitative counts were still correlated significantly with in-hospital mortality ($p = 0.02$ and $p = 0.005$ for Asian adults and children, and African children, respectively).

In the initial report demonstrating the prognostic value of pigment-containing PMN counts a 5% threshold was proposed for clinical practice[9]. We re-evaluated the prognostic value of this threshold in this larger patient population. Under a mixed-effects logistic regression model, combining individual patient data from the three randomised trials ($n = 6027$), greater than 5% pigment-containing PMNs was associated with an odds-ratio for death of 2.39 (95% CI 2.03–2.82). Greater than 5% pigment-containing monocytes ($n = 5819$) was associated with an odds-ratio for death of 1.31 (95% CI 1.10–1.55).

### Individual patient data meta-analysis of intraleukocytic malaria pigment counts

A systematic review of the literature identified five previously published studies which had recorded intraleukocytic pigment in over 100 patients with severe malaria[11,13,21–23]. We were able to obtain individual patient data from two of these studies[11,13]. We conducted an individual patient data meta-analysis using the pooled data set (overview given in Supplementary Table S1, $n = 32,038$ patients with data on the proportion of pigment-containing PMNs, $n = 31,010$ patients with data on the proportion of pigment-containing monocytes, and $n = 32,889$ patients with malaria parasite densities). There were no identified issues with data integrity in the pooled studies. A flow diagram for the inclusion and exclusion of the screened studies is given in Supplementary Fig. S3. All 5 studies included in the final individual patient data meta-analysis had a low risk of bias; the three studies in African children had moderate applicability due to poor specificity of the diagnosis of severe malaria in high transmission settings[20] (Supplementary Table S2).

**Prognostic value of quantitative microscopy counts in African children.** In our pooled data set, 30,868 African children had recorded parasite densities; 30,222 had recorded proportions of pigment-containing PMNs, and 29,377 had recorded proportions of pigment-containing monocytes. Figure 2 shows the estimated relationships between the quantitative microscopy counts and in-hospital mortality, using generalised additive regression models with random effect terms for each study and each study site. As observed in the AQUAMAT

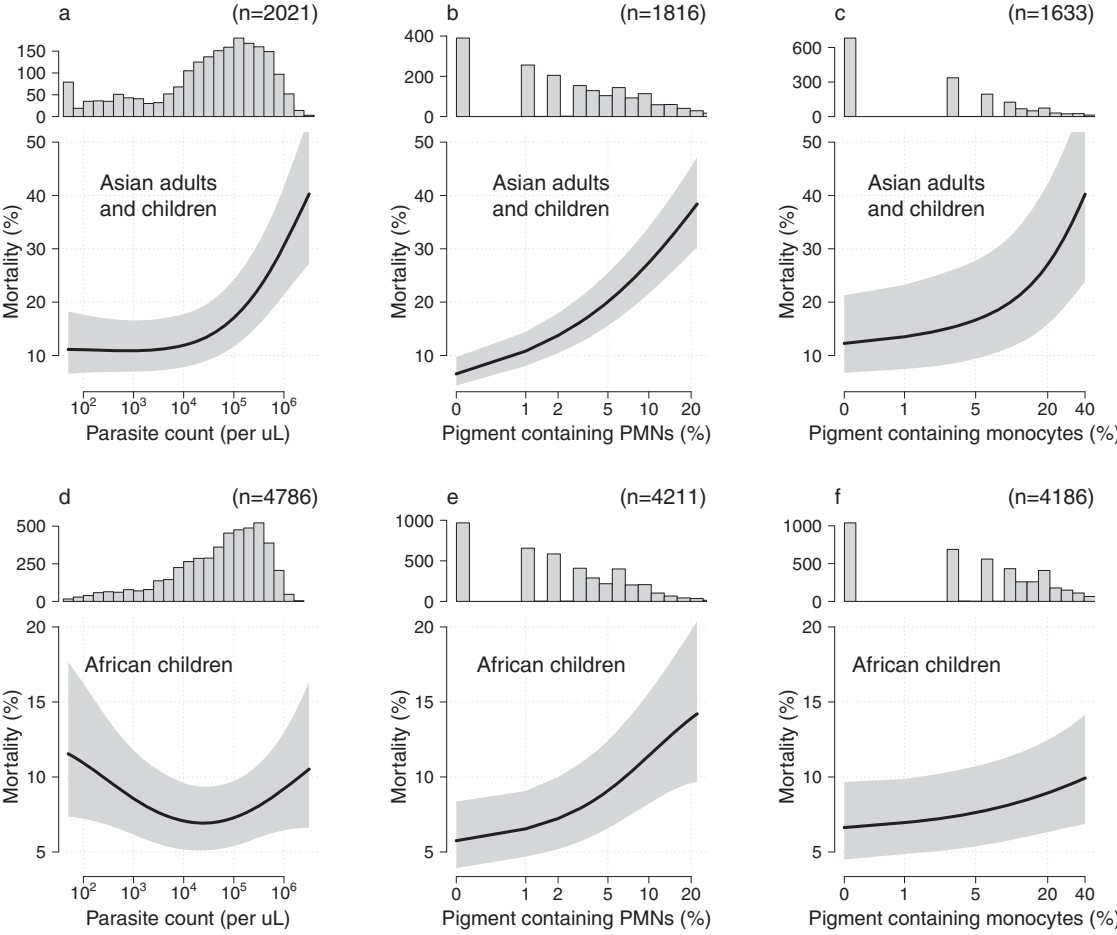

**Fig. 1 | New data from three randomised controlled trials in severe falciparum malaria.** The prognostic value of quantitative microscopy parasite counts, pigment containing neutrophil (PMN) counts and pigment containing monocyte counts. Panels **a**–**c** show data from Asian adults and children (AQ Vietnam and SEAQUAMAT trials); panels **d**–**f** show data from African children (AQUAMAT trial).

Each panel shows the histogram of quantitative counts (top), and the relationship between the count variable and mortality (mean: black line; 95% confidence interval: grey shaded area) estimated under a generalised additive logistic regression model (bottom). Note the y-axis scales are different for panels **a**–**c** and panels **d**–**f**.

trial, there was a 'U-shaped' relationship between the parasite count and a fatal outcome. Both low parasitaemia (<10,000 per µL) and high parasitaemia (>100,000 µL) were associated with an increased risk of death ($p = 10^{-16}$ for the spline fit). The proportions of pigment-containing PMNs and pigment-containing monocytes were both correlated positively with in-hospital mortality, with the greatest mortality increase observed for proportions of pigment-containing PMNs greater than 5% ($p = 10^{-40}$ and $p = 10^{-10}$ for the spline fits for the pigment-containing PMNs and monocytes, respectively).

**Prognostic value of >5% malaria pigment-containing neutrophils (PMNs).** In the pooled data set (Africa and Asia, $n = 32,035$), relative to patients with no pigment-containing PMNs, the greatest increase in mortality was seen in the subgroup of patients with >5% pigment-containing PMNs (odds ratio for death: 3.05 [95% CI 2.65–3.51]; Fig. 3a). The relationship between the pigment-containing monocyte counts and mortality was much less marked (Fig. 3b).

When comparing mortality outcomes in patients with >5% pigment-containing PMNs versus ≤5% pigment-containing PMNs, after pooling all available data, the meta-analytic estimate of the odds-ratio for in-hospital death was 2.63 (95% CI: 2.10–3.29, $p = 10^{-14}$, Fig. 4). In comparison, comparing mortality outcomes in patients with >5% pigment containing monocytes versus ≤5% pigment containing monocytes, the meta-analytic estimate of the odds-ratio for in-hospital death was 1.37 (95% CI: 1.11–1.67, $p = 10^{-3}$, Supplementary Fig. S4). Assuming a baseline risk of death of 5%, an odds-ratio for death of 2.6 translates to

a risk ratio of approximately 2.5, i.e. a mortality of 12.5% in patients with >5% pigment-containing PMNs compared to a mortality of 5% in patients with ≤5% pigment containing PMNs. In African children, this was a lesser prognosticator than coma (odds-ratio of 7.6) and acidosis (Kussmaul's breathing: odds-ratio of 6.5), but greater than severe anaemia (odds-ratio of 1.8) and high parasitaemia (odds-ratio of 1), see Supplementary Materials.

There was heterogeneity in the prognostic value of >5% pigment-containing PMNs when analysing adults versus children (>15 years vs ≤15 years). As apparent from Fig. 1, the prognostic value was greater in Asia (predominantly adults) as compared to Africa (all children). The odds-ratio for death in the pooled data set adults was 3.44 for adults (95% CI: 2.08–5.69, Supplementary Fig. S5), and 2.37 for children (95% CI: 2.37–5.99, Supplementary Fig. S6), a significant difference under an interaction model ($p = 0.002$).

**Prognostic value of intraleukocytic malaria pigment in addition to clinical and laboratory signs of severity**

The clinical utility of a prognostic test in severe malaria depends on whether it adds information to simple bedside assessment. With the exception of blood glucose, laboratory tests are not usually available immediately at point of care in the rural tropics -whereas most facilities do have a microscope (although unfortunately clean slides, good quality filtered stains and anhydrous methanol are often not available). We assessed the prognostic value of >5% pigment-containing neutrophils (PMNs) and >5% pigmented

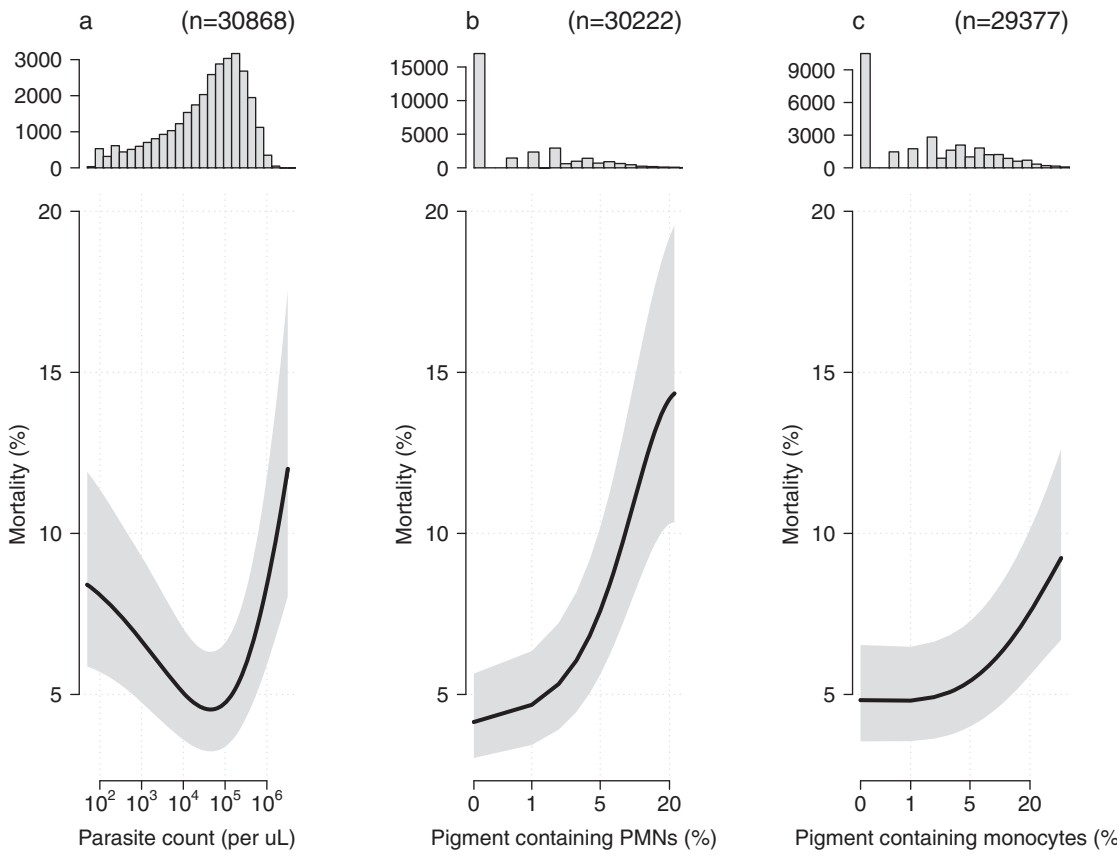

**Fig. 2 | The prognostic value of quantitative microscopy cell counts in the pooled data set of >30,000 children from sub-Saharan Africa diagnosed with severe falciparum malaria.** Each panel shows the histogram of quantitative counts (top), and the relationship between the count variable and mortality (mean: black line; 95% confidence interval: grey shaded area) estimated under a generalised additive logistic regression model (bottom). Panel **a**: parasite density; panel **b**: pigment containing neutrophils [PMNs]; panel **c**: pigment containing monocytes.

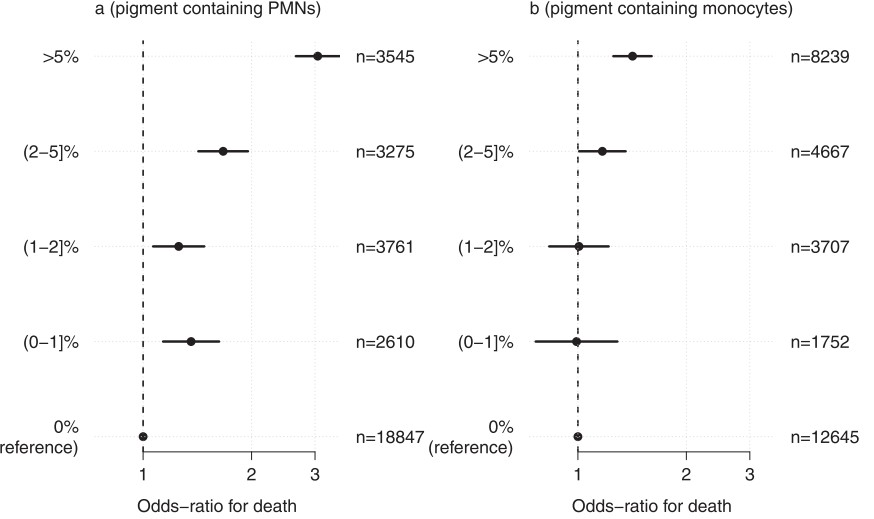

**Fig. 3 | Meta-analysis of the prognostic value of pigment-containing PMNs (polymorphonuclear leukocytes.** panel **a**) and monocytes (panel **b**) stratified into five levels with no pigment containing cells as the reference. Point estimates (filled circles) and 95% confidence intervals (thick lines) are shown, estimated under a logistic regression model with nested random effects for site, country and study.

monocytes before and after adjustment for readily measured known prognostic clinical and laboratory variables (Fig. 5). The clinical variables were coma (yes/no) and Kussmaul's breathing (for AQ Vietnam this was not recorded specifically so we used the respiratory rate as a proxy); the laboratory variables were hypoglycaemia and either lactate (for AQ Vietnam and SMAC) or blood urea nitrogen (for SEAQUAMAT and AQUAMAT). The study by Lyke et al. did not have any data on admission severity apart from coma so we omitted it from this analysis. The results show that pigment-containing PMN counts provided additional prognostic

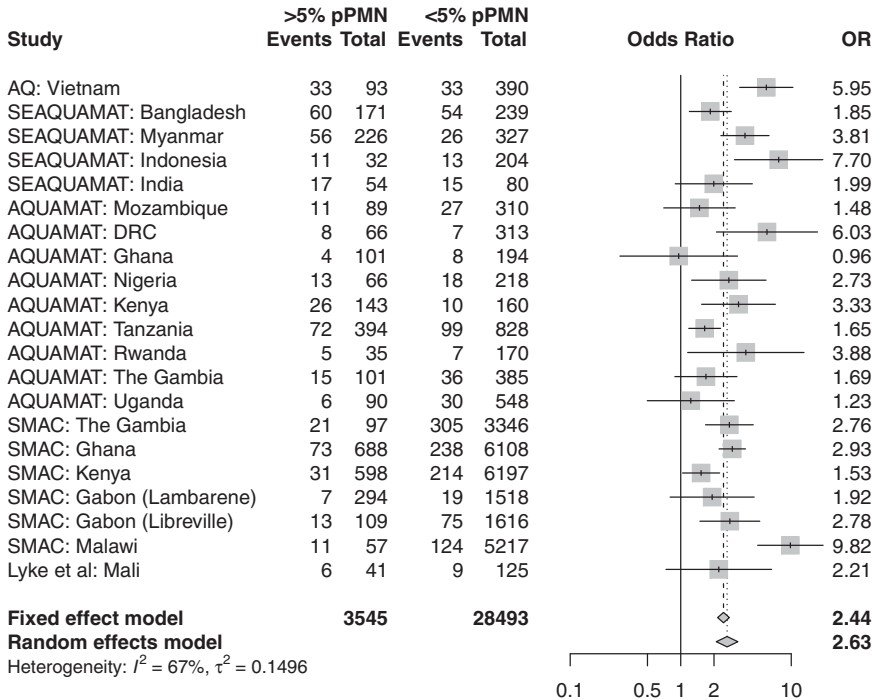

**Fig. 4 | Meta-analysis of the prognostic value for mortality of >5% versus ≤5% pigment-containing polymorphonuclear leukocytes (pPMN) in 32,035 patients clinically diagnosed with severe falciparum malaria.** The individual point estimates (shaded squares with centered vertical ticks) and 95% confidence intervals (horizontal lines) are shown for each study broken down by country of enrolment except for SMAC, Gabon where the distribution of pigment-containing PMNs was substantially different between the two sites (Lambaréné and Libreville).

information over that of the bedside clinical assessment and even laboratory biomarkers.

### Relationship between malaria parasite density and intraleukocytic pigment

We compared peripheral malaria parasite densities to the intraleukocytic pigment cell counts in the pooled data set (Supplementary Fig. S7). The peripheral parasite density was highly predictive of having >5% pigment-containing PMNs or monocytes (for a ten-fold increase in parasitaemia, the odds-ratio for having >5% pigmented PMNs was 2.4 [95% CI: 2.3–2.6], and the odds-ratio for having >5% pigment containing monocytes was 1.7 [95% CI: 1.6–1.7]). This translated roughly to less than 1 in 20 patients with parasite densities under 1000 per μL having pigment containing PMNs >5%, compared to 1 in 5 patients with high parasite counts (>100,000 per μL).

### Diagnostic value of intraleukocytic malaria pigment in severe illness

In African children distinguishing severe malaria from sepsis with incidental parasitaemia is difficult at presentation to the hospital or health centre[20,24]. Plasma *Pf*HRP2 is the best biomarker for the assessment of the sequestered parasite biomass in severe falciparum malaria (although the measurement is not widely available)[18]. The diagnosis of severe malaria is highly specific for patients with a plasma *Pf*HRP2 concentration ≥1000 ng/ml[20]. We assessed the diagnostic value of the proportion of pigment-containing PMNs and monocytes by comparing their values with the plasma *Pf*HRP2 concentrations in patients from the AQUAMAT trial for whom both were measured. We compared this with the peripheral parasite density. Plasma *Pf*HRP2 concentrations and proportions of pigment-containing PMNs and monocytes were jointly available for a total of 2933 and 2913 patients from the AQUAMAT trial, respectively. In total 1890 of these patients (64%) had a plasma *Pf*HRP2 concentration above 1000 ng/ml. The proportions of pigment-containing PMNs and pigment containing

monocytes were both highly predictive of having a plasma *Pf*HRP2 concentration ≥1000 ng/mL (Fig. 6, $p = 10^{-26}$ and $p = 10^{-15}$ for a mixed-effects logistic regression model fits, respectively). In patients with ≤5% pigment-containing PMNs, 1306 out of 2178 (60%) had a plasma *Pf*HRP2 ≥1000 ng/mL. In comparison, in patients with >5% pigment-containing PMNs, 583 out of 755 (77%) had a plasma *Pf*HRP2 ≥1000 ng/mL. The corresponding proportions with less than and greater than 5% pigment-containing monocytes were 672 out of 1208 (56%) and 1204 out of 1705 (71%), respectively. In comparison, only parasite counts above 100,000 per μL were predictive of having a plasma *Pf*HRP2 ≥1000 ng/mL (Fig. 6).

### Discussion

Severe falciparum malaria is estimated to kill over 1000 people each day. Most of these preventable deaths are in young children in Africa. Rapid recognition of the disease and early administration of effective treatment are critical determinants of outcome[2,16,17,25,26]. Delays in diagnosis and the administration of artesunate cost lives[1,2]. In hospitals and health centres a diagnosis of malaria and recognition of the three major clinical components of severe malaria (coma, acidotic breathing, severe anaemia) can be done immediately but, importantly, this does not distinguish malaria as the cause of severe illness, from other causes of severe febrile illness (e.g. sepsis) and incidental malaria. The diagnosis of malaria has traditionally been made by microscopy of stained blood films. However today, rapid diagnostic tests (RDTs) are increasingly used to diagnose malaria, even in hospitals, and there is less use of microscopy than in previous years. There are concerns that microscopy proficiency and support are declining in malaria endemic areas[1], yet microscopy of thin and thick blood films provides important diagnostic and prognostic information which the current rapid tests do not. Blood slide microscopy gives quantitative information which is not vulnerable to technical errors or to mutations in the parasite target genes (e.g. *Pf*HRP2 mutations, which can cause false RDT negativity[27,28]). Thin blood films have the advantage over thick films of

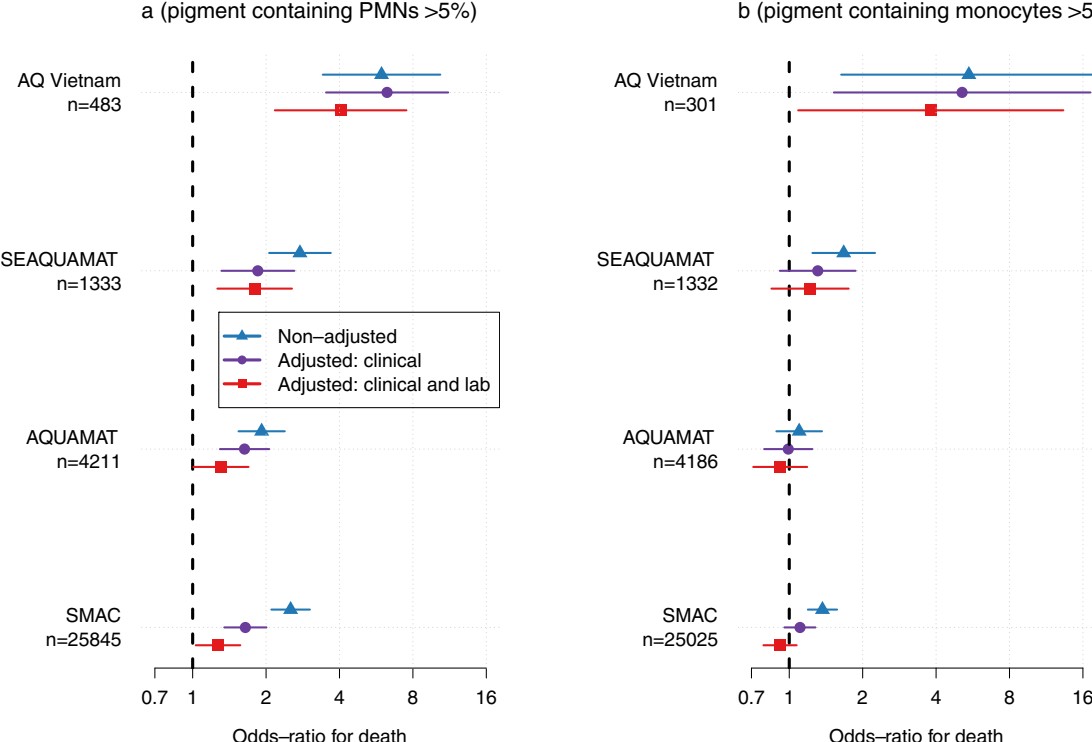

**Fig. 5 | The prognostic value of intraleukocytic pigment cell counts in severe falciparum malaria with and without adjustment for known prognostic clinical and laboratory variables (clinical: coma and acidosis; laboratory: hypoglycaemia and either venous lactate for AQ Vietnam and SMAC or blood urea nitrogen for SEAQUAMAT and AQUAMAT).** Panel **a**: PMNs (polymorphonuclear leukocytes); panel **b**: monocytes. The point estimates (95% confidence intervals) for the nonadjusted models are shown by the blue triangles (blue lines); for the models adjusted with clinical variables only are shown by the purple circles (purple lines); for the models adjusted with clinical and laboratory variables are shown by the red squares (red lines).

speed and ease of assessment (see[29]) and they provide valuable prognostic information. In addition to the malaria parasite count (the proportion of red cells containing one or more parasites), the stage of asexual parasite development can be assessed[8] and the proportion of leukocytes containing ingested malaria pigment in the blood slide can be counted[9]. Polymorphonuclear leukocytes (PMNs: granulocytes; nearly all neutrophils) are more abundant than monocytes, are readily identified, and an assessment of 100 PMNs in the tail of the thin blood film takes only a few minutes (see[29]). To evaluate the diagnostic and prognostic value of counting the proportion of pigment-containing PMNs this study used prospective assessments from three of the largest randomized controlled trials in severe falciparum malaria, and then combined these within an individual patient data meta-analysis of all the large studies which recorded intraleukocytic pigment data. In total this combined data set comprises the majority of all patients with severe malaria who have been studied prospectively in the past half century. The very large data set shows clearly the prognostic and diagnostic value of counting the proportion of PMNs with intraleukocytic malaria pigment. It is a simple and rapid assessment which provides additional prognostic information above that provided by a bedside examination, even when it is supplemented by other point-of-care tests.

The utility of counting the proportion of pigment containing PMNs in patients with suspected severe malaria has to be understood in the context of delivering critical care in busy resource-limited hospitals and health centres. In higher transmission settings differentiating severe malaria from other causes of severe childhood febrile illness is difficult, even for experienced clinicians, because the key clinical signs are not specific, and asymptomatic malaria parasitaemia is so common. Thus, many children are diagnosed as having severe malaria, whereas in fact they have another severe illness (usually bacterial sepsis[24] with incidental parasitaemia)[19,20,30–32]. Finding a high

proportion of pigment-containing PMNs supports the diagnosis of severe malaria and it identifies those patients in need of intensive care or, if this is unavailable, close monitoring. Prognosis and diagnosis are linked: if malaria is not the cause of severe illness, then potential sources of sepsis need to be investigated and parenteral antibiotics need administering rapidly[32]. This dilution by a different cause of fatal illness explains why the prognostic value of the intraleukocytic malaria pigment counts in African children is less than in Asian adults and children (Fig. 2). In the latter group the specificity of malaria parasitaemia as an indicator that malaria is the cause of severe illness is substantially higher. However, the proportion of PMNs containing malaria pigment is a better diagnostic and prognostic indicator than the commonly reported malaria parasite count. Overall the peripheral blood parasite density is a poor prognosticator in African children with severe malaria[5] because there is a 'U-shaped' relationship with in-hospital mortality and because it is a poor measure of the sequestered parasite burden[18]. Although it is recommended that both thin and thick blood films are taken from patients suspected of having severe malaria, in many centres only a rapid test or a thick blood film is performed[33]. Quality thin films require clean glass slides, filtered stains, anhydrous methanol, and good technique (see[29]). But they are much easier to read, they provide more accurate counts in high-density infections, and they allow more accurate staging of parasite development, and counting of pigment-containing leukocytes. Thick films take time to dry, and are more difficult to evaluate accurately because cellular morphology is contracted, and artefacts and stain precipitates may be mistaken for malaria pigment. The speed and simplicity of counting pigment-containing PMNs is particularly valuable in rural settings with limited access to laboratory support, and thus limited capability of providing other prognostic information. Health systems should support maintenance of quality microscopy. New inexpensive cell phone-adapted automated methods of microscopy cell counting

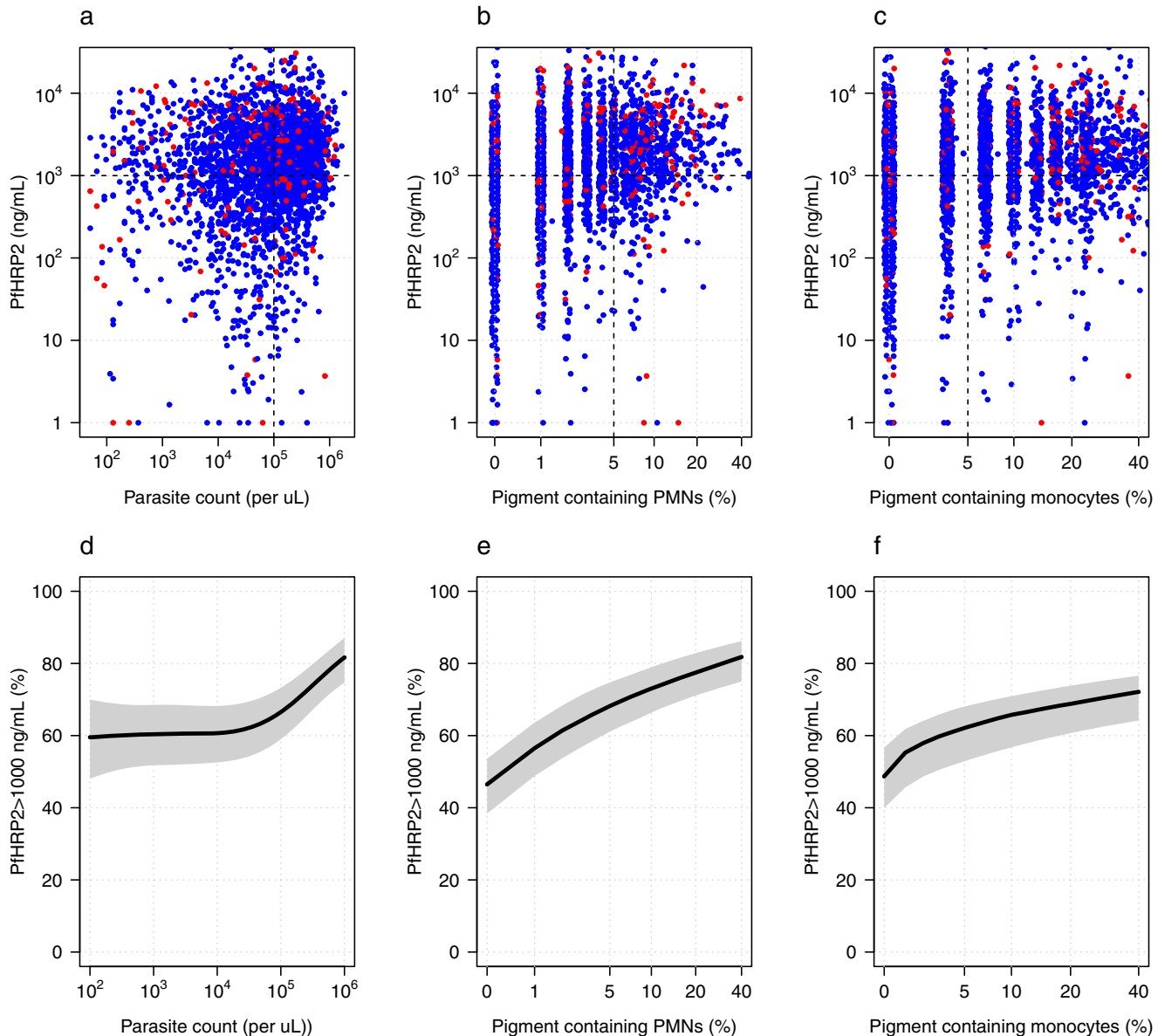

**Fig. 6 | The diagnostic value of the parasite count, pigment-containing PMNs (polymorphonuclear leukocytes) and monocytes in severe falciparum malaria.** Panels **a**–**c** show scatterplots of the parasite count and the proportions of pigment containing PMNs and monocytes, respectively, against the plasma *Pf*HRP2 concentrations in the 2933 patients from the AQUAMAT trial who had all measurements recorded. Red: patients who died; blue: patients who survived. In panels **d**–**f**, the black lines (grey shaded areas) show the mean (95% confidence interval) probability of having a plasma *Pf*HRP2 concentration ≥1000 ng/ml as a function of the parasite count and the pigment containing leukocyte counts.

are now being developed and evaluated. These may be adaptable to parasite counting, staging and neutrophil pigment assessment but they are still years from being implementable in remote areas[34]. Where there is a microscope, thin film assessment should be a routine part of the assessment of any patient suspected of having severe falciparum malaria.

Sequestration of parasitised erythrocytes in the venules and capillaries of vital organs is the fundamental pathophysiological process in severe falciparum malaria[30,35]. The concentrations of *Pf*HRP2 and parasite DNA in plasma are quantitative measures of the preceding hidden sequestered parasite burden as both are liberated into the circulating plasma at schizont rupture. Both measurements correlate strongly with disease severity[18,36]. The obstructed microcirculation in the retinal, buccal or rectal circulations can be visualized in-vivo[37]. Indirect ophthalmoscopy by an experienced clinician can identify characteristic retinal and vascular changes which are highly specific for falciparum malaria as the cause of coma[30]. However both these approaches require special training and relevant equipment at the bedside, and they are generally not performed. Quantifying the proportion of pigment-containing neutrophils on a thin blood film is a simple, rapid, inexpensive, and generalizable alternative way to assess the preceding sequestered parasite biomass. *Pf*HRP2 remains in the bloodstream for days, and so it accumulates with successive schizogony, whereas *P. falciparum* DNA and haemozoin are removed more rapidly from the circulation. The rate of removal of haemozoin depends on the turnover and clearance of the cells which have phagocytosed it. PMNs have more rapid turnover than monocytes, with a half-life in the circulation of about 19 hours compared to 45 hours for monocytes[12]. PMN blood counts in severe malaria are usually normal or slightly raised[38]. Thus, the proportion of PMNs containing intraleukocytic pigment is a function of the amount of recent schizogony. This explains its prognostic value. By contrast monocytes which have

ingested pigment reflect a longer time course, which is why they are associated with anaemia[39]. The number of monocytes with intraleukocytic malaria pigment still has prognostic value in severe malaria, but much less so than for PMNs. The blood cell counts also inform the diagnosis. Finding PMNs with intraleukocytic malaria pigment points to malaria as the cause of severe illness, whereas higher platelet counts, higher white blood cell counts, absence of PMNs containing pigment and low parasite counts all point towards an alternative cause of severe illness[19,20,24].

The principal limitation of this study is the inaccuracy in the diagnosis of severe malaria in children in areas of higher transmission[20]. As the mortality of sepsis is higher than that of severe malaria, misdiagnosis dilutes the estimate of the prognostic value of true malaria associated severity indicators. The definition of severe malaria also varied between studies, with strict definitions in the randomised controlled trials and looser more inclusive definitions in the observational studies -notably the large SMAC study[13]. As the data reported in this study represent the majority of all available relevant patient data, selection or reporting biases are unlikely to be significant confounders. Furthermore, as counting was performed by microscopists unaware of the patient's condition or outcome, counting biases are also very unlikely. Counts in thick blood films, as in the large SMAC study, are less accurate than thin film counts but the effect sizes observed were not markedly different to those series in which thin film counts were performed.

Our conclusions differ from those of the SMAC network analysis[13]. Their data analysis concluded that "although high levels of pigmented cells were associated with a fatal outcome in some study sites, they were not useful predictors of outcome across Africa" and that "they were not useful markers of fatal outcome for individual patients". Although the children studied in SMAC had a much broader range of disease severities (i.e. in contrast to the randomised controlled trials evaluated here, a substantial proportion of children in the SMAC database likely did not have severe malaria and the overall mortality was low), and thick blood films rather than thin films were evaluated, reanalysis of the SMAC data shows clear prognostic value of intraleukocytic pigment, consistent with the four other malaria studies (Supplementary Fig. S8). In summary, the proportion of PMNs containing malaria pigment has diagnostic and prognostic value, and importantly it adds to the bedside assessment of coma and acidotic breathing.

## Methods
All clinical trial protocols were reviewed and approved by each site's appropriate ethical review board (ERBs), and also by the Oxford Tropical Research Ethics committee (OXTREC). Re-use of existing, appropriately anonymized, human data does not require ethical approval under the Oxford Tropical Research Ethics Committee regulations (OxTREC).

### Clinical trials
This work presents new data from three previously published randomised clinical trials in severe malaria. The three clinical trials were conducted by associated research teams. The AQ Vietnam trial was a very detailed single-centre study in Vietnamese adults with severe falciparum malaria[15]. Patients were included in the study if they (or an accompanying relative) gave informed consent, if the patient had asexual forms of *P. falciparum* on a peripheral-blood smear, was older than 14 years, was not in the first trimester of pregnancy, was not an intravenous drug user, and had received less than 3g of quinine or two doses of artemisinin or a derivative in the previous 48 hours. The severe malaria entry criteria were modified from the WHO criteria as follows: patients could be included if they had one or more of the following: a Glasgow Coma Scale score of less than 11 (indicating cerebral malaria); anaemia (haematocrit <20%) together with a parasite count >100,000/μL; jaundice (serum bilirubin >2.5 mg/dL [50 μmol/L]) together with a parasite count >100,000/μL; renal impairment (urine output, 400 ml per 24 hours; and serum creatinine > 3 mg/dL [250 μmol/L]); hypoglycaemia (blood glucose < 40 mg/dL [2.2 mmol/L]); hyperparasitaemia (>10% parasitaemia); and systolic blood pressure <80 mm Hg with cool extremities (indicating shock). Intensive clinical and laboratory monitoring was performed by a specialist team in a dedicated ward. This trial provided the basis for the design and conduct of the multi-centre trials conducted in adults and children with severe falciparum malaria in South-East Asia (SEAQUAMAT: mainly adults[16]), and then in Africa (AQUAMAT: mainly children[17]). The AQ Vietnam and SEAQUAMAT trials were conducted before pre-registration was standard practice; the AQUAMAT trial was registered on ISRCTN (ISRCTN50258054).

**AQ Vietnam trial.** The AQ Vietnam trial was a double-blind randomised controlled comparison of intramuscular artemether and intramuscular quinine in 560 Vietnamese adults with strictly defined severe falciparum malaria. It was conducted between May 1991 and January 1996 in a specialist ward of the Hospital for Tropical Diseases, Ho Chi Minh City. Malaria blood film microscopy was performed on admission before enrolment. Full details have been published previously[15], and data on the prognostic value of pigment-containing PMN and monocyte counts were published for the first 300 patients[9].

**South East Asian Quinine Artesunate Malaria Trial (SEAQUAMAT).** The SEAQUAMAT trial was a multicentre open label randomised comparison of parenteral artesunate and parenteral quinine in 1461 Asian adults and children admitted to hospital with severe falciparum malaria[16]. The participating centres were located in Bangladesh, Myanmar (7 hospitals), India, and Indonesia. Patients over two years of age were included in the study if they had a positive RDT for *P. falciparum* histidine rich protein 2 (*Pf*HRP2; Paracheck: Orchid Biosystems, Goa, India) and, in the admitting physician's clinical opinion, they had severe malaria, and they or their attendant relative of guardian, gave fully informed written consent. Patients were excluded if there was a convincing history of full treatment with quinine or an artemisinin derivative for more than 24 hours before admission. Full details of the trial have been published previously[16].

**The African Quinine Artesunate Malaria Treatment Trial (AQUAMAT).** The AQUAMAT trial was a multicentre open-label randomised comparison of parenteral artesunate and parenteral quinine in 5425 African children admitted to hospital with severe falciparum malaria[17] (ISRCTN: 50258054). The eleven participating centres in nine countries across Africa were: Hospital Central da Beira, Beira, Mozambique; Royal Victoria Teaching Hospital, Banjul, The Gambia; Komfo Anokye Hospital, Kumasi, Ghana; Kilifi District General Hospital, Kilifi, Kenya; Magunga District Hospital, Korogwe, Tanzania; Teule District Hospital, Muheza, Tanzania; University of Ilorin Teaching Hospital, Ilorin, Nigeria; Mbarara Teaching Hospital, Mbarara, Uganda; Kingasani Health Centre, Kinshasa, DRC; Rwamagana Hospital, Rwamagana, Rwanda and Nyanza Hospital, Nyanza, Rwanda. Children less than 15 years were included in the study if they had a positive RDT for *Pf*LDH (Optimal®) and, in the admitting physician's clinical opinion, they had severe malaria, and they or their attendant relative or guardian gave fully informed written consent. Thus the inclusion criteria of a clinical diagnosis of severe malaria (guided by the WHO criteria) by the admitting clinician were similar in AQUAMAT and SEAQUAMAT trials. Patients were excluded if there was a convincing history of full treatment with quinine or an artemisinin derivative for more than 24 hours before admission. Overall a third of enrolled patients had cerebral malaria. Full details of the trial have been published previously[17].

## Procedures

In the three trials, an initial clinical assessment was recorded and a peripheral thick and thin blood smear was made for quantitative malaria parasite counting. In the AQ Vietnam study, blood was also taken for immediate haematocrit, blood glucose, plasma lactate and blood gas measurements, blood culture, cross match if necessary, and same-day laboratory haematology and biochemistry. Pigment containing leukocytes on the admission blood films were counted later by expert microscopists blinded to the patient outcomes. In the SEAQUAMAT and AQUAMAT trials, blood was taken for immediate haematocrit and biochemical analysis using the EC8+ card in a hand held battery operated internally calibrated biochemical analyser (i-STAT, Abbott Laboratories, Il, USA). This provided an immediate hard copy readout with time and date. Parasitised erythrocytes were counted on the Giemsa or reverse Field's stained thin film and reported per 1000 red cells, or if the count was low or the thin film unavailable, was counted in the thick film (and reported as parasites/200WBC). Parasite counts were checked, and parasite staging and leukocyte-associated pigment was assessed at the reference laboratory in Bangkok. In each study the numbers of leukocytes containing visible malaria pigment were counted per 100 PMNs and per 30 monocytes under oil immersion microscopy at X1000 magnification on thin blood films (or thick films if the thin film was unavailable).

A video showing how to perform a blood smear (for thick and thin blood films), as well as how to assess slide quality, and count intraleukocytic malaria pigment is provided at https://doi.org/10.6084/m9.figshare.20783683.v1[29].

## Systematic review

We searched PubMed and EMBASE for the terms ('intraleukocytic pigment' OR 'pigment' OR haemozoin) AND ('severe malaria') to find all studies that measured the proportions of either pigment-containing PMNs or pigment-containing monocytes in patients diagnosed with severe falciparum malaria. In addition we reviewed all studies included in a recent systematic review of prognostic factors in African children with severe malaria[6], and searched Google Scholar for publications that cited the original report on the prognostic value of intraleukocytic pigment in severe malaria (Phu et al.)[9]. Studies were screened for eligibility by two reviewers (JAW and KS) and any disagreements were adjudicated by a third (NJW).

Out of a total of 202 publications identified, after removing duplicates, reviews, case-reports, animal studies, studies in nonsevere disease and non-*P. falciparum* malaria, we identified 19 relevant studies. The median sample size was 67 with a range of 22 to 26,296. Summary details and findings for each study is provided in the Supplementary Materials.

We contacted investigators of the 5 studies which had data on at least 100 patients diagnosed with severe malaria[11,13,21–23]. For three of the studies we were unable to obtain the individual patient data: for Boeuf et al.[22] the P.I was deceased; for Luty et al.[21] the original data could not be found by the authors; for Birhanu et al.[23] the contact details were out of date and we failed to contact the corresponding author. The systematic review was registered prospectively on PROSPERO, number CRD42021284527 (https://www.crd.york.ac.uk/prospero/display_record.php?RecordID=284527).

## Additional studies from systematic review

**1. Mali study: Lyke et al.** This was a prospective study of severe malaria, designed as a pilot for the larger SMAC network study. 172 patients diagnosed with severe malaria were enrolled in Bandiagara, Mali from July 2000 to December 2001[11]. The proportions of pigment containing PMNs and monocytes were quantitated on thin films by counting 100 PMNs and 30 monocytes, respectively. Microscopists were blinded to clinical presentation and outcome. 20% of patients had coma and 15 patients died (9%).

**2. Severe Malaria in African Children (SMAC).** SMAC was a prospective multicentre network observational study[33]. The evaluated population comprised all parasitaemic children suspected of having severe malaria who were admitted to each of the participating hospitals of the SMAC network. The SMAC network included sites in five African countries: Banjul, The Gambia (Medical Research Council Laboratories, Malaria Research Programme, in collaboration with the Royal Victoria Teaching Hospital); Blantyre, Malawi (Blantyre Malaria Project, Queen Elizabeth Central Hospital); Kumasi, Ghana (University of Science and Technology, School of Medical Science); Kilifi, Kenya (Kenya Medical Research Institute for Geographic Medicine); and two sites in Gabon, Lambarene and Libreville, both run by the Medical Research Unit of Albert Schweitzer Hospital. All children aged between 1 month and 15 years of age suspected of having malaria and who were sick enough to be hospitalized were screened with a thick blood smear for *P. falciparum* parasites. Only a minority of the children in the SMAC study had strictly defined severe malaria[2]. Clinical and laboratory measures were documented and children were followed during their hospital admission. Pigment-containing mononuclear leukocytes, and pigment-containing granulocytes (PMNs) were counted as described by Lell et al. in stained thick blood smears[40]. The numbers of pigment-containing monocytes and pigment-containing granulocytes were reported per 200 cells. Between December 2000 and May 2005 26,389 patients were enrolled. Overall 9.3% of enrolled patients had coma (BCS ≤ 2) and 4.4% died. Full details have been published previously[13].

**Quality assessment.** The risk of bias and the applicability of the 5 included studies was assessed by applying the 2011 revised version of the Quality Assessment of Diagnostic Accuracy Studies (QUADAS-2) tool[41]. We adjusted QUADAS-2 so that it was relevant to the key questions of interest using 5 domain questions (risk of bias: patient selection, microscopy evaluation, outcome evaluation; applicability: patient selection and outcome evaluation). Quality assessment was performed by two reviewers (JAW, KS). Any disagreements were resolved by a third reviewer (NJW).

## Statistical analysis

**Causal structure.** The analyses were guided by the simplified causal diagram shown in Supplementary Fig. S9. The primary determinant of mortality in severe malaria is the total parasite biomass (or more exactly the proportion of parasitised red cells). This is unobservable due to sequestration of later parasite stages. Intraleukocytic pigment counts, plasma *Pf*HRP2, and the peripheral parasite density are proxy measurements for the parasite biomass of varying accuracy, and thus are expected to have prognostic utility[18,20]. Clinical measurements such as coma are best defined as mediators, i.e. the effect of parasite biomass on mortality is due to organ dysfunction (which can be measured to varying degrees of accuracy). Thus the prognostic value of indicators of parasite biomass such as intraleukocytic pigment after adjustment for clinical variables can be thought of as the direct effect (not going via the mediators).

## Prognostic value of quantitative microscopy cell counts

**Primary endpoint.** In all analyses we used in-hospital mortality as the primary endpoint (this was the primary endpoint in all three randomised trials). We used the log-odds scale to estimate effects. Only patients with available and readable slides were included in these analyses.

**Exposure.** The primary exposure definition was > 5% pigment containing PMNs or > 5% pigment containing monocytes. The 5% threshold was proposed in the first paper noting the prognostic value of PMN and monocyte counts[9]. In addition, it is an easy to remember threshold and thus appropriate for use in clinical practice.

Dichotomising continuous variables loses key information. We also performed analyses using the continuous count data. For all the microscopy cell counts, there were a substantial number of zero counts, and the distributions were highly right skewed as a consequence. We used a Box-Cox $\log_{10}(x+\lambda)$ transformation to reduce the right skew whilst accounting for the zero counts where, for the pigment containing PMNS and monocytes, $x$ is expressed as a percentage and $\lambda = 0.5$; and for the parasite counts $x$ is the per $\mu$L density and $\lambda = 50$ (approximately half the lower limit of detection for a thick film). Sensitivity analyses varied these $\lambda$ parameters. For the proportions of pigment containing PMNs and monocytes, estimated odds-ratios for death are sensitive to the choice of the additive parameter in this Box-Cox transformation. The value of $\lambda = 0.5$ was chosen as this resulted in approximately the same odds-ratio when analysing all data under the $\log_{10}(x+\lambda)$ transformation, or only the positive count data under a $\log_{10}$ transformation.

**Analysis population.** For each included study, the analysis population was all patients who had available and readable slides (no imputation of missing count data).

**Analyses.** To estimate the relationship between baseline microscopy quantitative cell counts and in-hospital mortality, we fitted univariable generalised additive logistic regression models with mortality as the dependent (outcome) variable, and the baseline counts on the $\log_{10}$ scale as the dependent (predictor) variable.

The generalised additive logistic regression models were fit using the R package *mgcv* which implements penalised regression using smooth splines. We chose a spline based regression approach as the association between mortality and baseline quantitative microscopy cell counts could plausibly be non-linear on the logit scale. To avoid over-fitting we set the maximum basis for the smooth spline component to be 4. Random effect terms were included for country of enrolment and study. Independent models were fitted to the data from Asian adults and children (SEAQUAMAT and AQ Vietnam), and African children (AQUAMAT). In addition we fitted models to all available data from African children (AQUAMAT, SMAC and the Mali study).

**Prognostic models adjusted for baseline severity.** To characterise the prognostic utility of malaria pigment-containing leukocyte count data after adjustment for key clinical variables, we fitted additional prognostic models of death adjusted for known prognostic clinical variables (coma: yes/no; suspected acidosis: yes/no) or known clinical and readily measured laboratory variables (hypoglycaemia: yes/no; blood urea nitrogen; lactate)[5].

Coma was defined in children as a Blantyre coma score (BCS) ≤2 and in adults as a Glasgow coma score (GCS) ≤10. Suspected acidosis was defined clinically in the SEAQUAMAT and AQUAMAT trials as acidotic breathing and in SMAC as deep breathing (Kussmaul respirations). In the AQ Vietnam study this was not recorded explicitly so we used the proxy variable of respiratory rate. Hypoglycaemia was defined as a blood glucose ≤2.2mmol/L. Lactate was measured in venous plasma for the AQ Vietnam trial and in venous whole blood in the SMAC study. Blood urea nitrogen in SEAQUAMAT and AQUAMAT was measured with the i-stat handheld analyser. The prognostic models were logistic regression models with the binary predictor of >5% pigment-containing PMNs or monocytes and with nested random intercept terms for hospital, country and study, fitted using the R package *lme4*.

**Meta-analysis.** To estimate the meta-analytic prognostic value of a pigment containing PMN count or a pigment containing monocyte count ≥5% with respect to in-hospital mortality, we fitted a logistic regression model with nested random effects by hospital, country and

study (for SMAC we had data only at the country level except for Gabon). We looked at heterogeneity in effect between adults versus children (<15 years versus >15 years) by adding an interaction term in the random effects logistic regression model. For children we also fitted generalised additive logistic regression model (R package *mgcv*) to the transformed count data (see above section on Box-Cox transformations used). Estimation and forest plot visualisation of the meta-analysis was done with the R package *meta*.

**Diagnostic value of intraleukocytic pigment counts.** In previous work we assessed the accuracy of the clinical diagnosis of severe malaria in large patient cohorts from Africa and Asia[19,20]. A major finding was that approximately one third of children in high transmission settings likely had another cause of severe disease, most likely bacterial sepsis[24]. Plasma *Pf*HRP2 is the best available diagnostic marker for severe malaria[18,20]; a lower threshold value of 1000 ng/ml has approximately 74% sensitivity and 93% specificity[20]. As there is no 'gold-standard' diagnosis against which to assess candidate diagnostic markers, estimates of misdiagnosis used latent class models, jointly modelling platelet counts and plasma *Pf*HRP2. Platelet counts were not recorded in AQUAMAT. Intraleukocytic pigment and plasma *Pf*HRP2 are measuring the same underlying quantity (total parasite biomass), and thus latent class model fits are highly sensitive to model assumptions because of the within-class correlation. For this reason, to assess diagnostic utility we simply assessed the predictive value of intraleukocytic pigment for the plasma *Pf*HPR2 (under the causal diagram in Supplementary Fig. S9 this correlation results from both variables being proxies of the underlying parasite biomass). We fit logistic regression models to the dichotomised outcome variable (plasma *Pf*HPR2 > 1000 ng/ml) with random effects for each study site. We used a bootstrap approach to calculate confidence intervals around the fixed-effect predictions in mixed effects models (R package *bootpredictlme4*).

### Ethics & Inclusion statement

This research has included local researchers throughout: in the study design, study implementation, data ownership, and authorship of publications. This research has practical and implementable consequences for the treatment of severe disease in the included areas. All clinical studies were approved by local ethics review committees.

### Reporting summary

Further information on research design is available in the Nature Portfolio Reporting Summary linked to this article.

## Data availability

Pigment containing PMN and monocyte counts for the AQ Vietnam, SEAQUAMAT and AQUAMAT studies are available along with parasite counts, coma, acidosis and outcome on the github repository which has been deposited on Zenodo (DOI: 10.5281/zenodo.5720162)[42]. Data from the SMAC network are openly available at: https://dataverse.harvard.edu/dataset.xhtml?persistentId=doi:10.7910/DVN/0CTWUJ.

## Code availability

All the code and data are available via a github repository: https://github.com/jwatowatson/MalariaPigmentPrognosis.

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

## Acknowledgements

We are very grateful to our many colleagues who conducted the AQ Vietnam, SEAQUAMAT and AQUAMAT trials and to the SMAC network who generously made their anonymised study data openly accessible. This research was funded, in whole or in part, by The Wellcome Trust. A CC BY or equivalent licence is applied to the author accepted manuscript arising from this submission, in accordance with the grant's open access conditions. NJW is a Principal Research Fellow funded by the

Wellcome Trust (093956/Z/10/C). JAW is a Sir Henry Dale Fellow funded by the Wellcome Trust (223253/Z/21/Z). We thank Supaat Asarath (Ice), Thanawat Assawariyathipat, Ranitha Vongpromek and Mehul Dhorda for making the accompanying tutorial video on how to perform blood smears for thick and thin blood films with a demonstration of how to count intraleukocytic pigment.

## Author contributions

J.A.W. and N.J.W. conceived the study. N.J.W., N.P.J.D., A.M.D., Lv.S., K.E.L., P.T.D., N.H.P. and C.F. designed and/or conducted the clinical trials. Ketsanee S., B.I., Kalmorat S. and K.C. did the microscopy. Ketsanee S., J.A.W. and N.J.W. reviewed all papers from the systematic review. J.A.W. analysed the data and wrote the first draft of the manuscript. N.J.W. supervised the study. All authors read, edited and approved the final manuscript.

## Competing interests

The authors have no competing interests.
