## [Peer review file · Nature Communications]

REVIEWER COMMENTS

Reviewer #1 (Remarks to the Author):

The paper seeks to show that the proportion of pigment containing polymorphonuclear leukocytes in peripheral blood films is strongly positively correlated with prognosis in malaria. Whilst for those of us working in malaria diagnostics the motivation for the study is clear, there is currently a lack of clarity over what is being claimed, with conclusions drawn often being qualified.

It is probable that the paper would be of interest to those with a specialist knowledge of malaria testing. As written, the messages within the text do not provide clear unambiguous results that would be of interest to a broader readership associated with this journal. It is an output that may sit more naturally in a specialist journal such as *Malaria*, where the nuances of the messaging might be better understood.

There are a number of technical clarifications that should be addressed by the authors.

1. The study selection process is, as written, unclear. The sourcing of the seemingly pre-defined AQ Vietnam, SEAQUAMAT and AQUAMAT, in relation to the systematic review studies (SMAC, Lyke the al. and Bouef et al.) should be clarified. This raises a concern around the comprehensive sampling of the evidence-base. The authors should consider replicating their search on other databases (at least one other of MEDLINE/ EMBASE/ etc.).
2. The methodology of the literature screening is not clearly reported - this should ideally be replicated by 1 or more authors and the process should be summarised with a PRISMA flow diagram, as well as in the Nature Communication study protocol.
3. A concern arises around the meta-analysis populations. Firstly, although inclusion and exclusion criteria are stated clearly in the review, importantly, SMAC is a retrospective analysis. This may be at odds with the inclusion criteria sought, as registered on Prospero, thus challenging the conduct of the review. Differences between studies also introduce concerns around the internal validity of the analysis. Inclusion and exclusion criteria for each study are not clearly stated, nor is how the analysis was adapted for missing patient data (especially PMN/ PMM counts).
4. Focussing specifically on the SMAC analysis, this study is both significantly larger (n=26,389) and presents a patient population with less severe malaria (4% mortality, 9% coma). Although in the

discussion the authors state that they conducted their analysis on a severe malaria subgroup, the criteria and baseline characteristics for this group are not clarified in the text. Given that this is used to evidence the validity of the prognostic indication of >5% pPMN, it may raise concerns around selection bias in this cohort. Additionally, the smaller subgroup may indicate a smaller analysed population than the title indicates.

5. A further concern arises in the geographic heterogeneity between Asian and African patient populations and whether this confounds the meta-analysis conducted. There may be a risk of substantial heterogeneity in Figure 3 ($I^2= 68\%$), which further evidences this; although, this is not commented upon by the authors nor does there seem to be any methodological contingencies to explore this: pre-specified or ad-hoc. Similarly, there is clear inconsistency in the age of patients between studies. This is important as immunological maturation may influence malarial disease course and sequelae, possibly further confounding the internal validity of this analysis.

6. Noticeably, there appears to be no risk of bias assessments (or references to such) at either the study or outcome levels of the review. This may indicate that the authors have not critically considered sources of variation or error in their prognostic modelling and challenges the validity of the work they have undertaken. For each study included, a risk of bias evaluation (ideally with an externally validated tool) should have been conducted for each outcome extracted and, additionally, where synthesised, a funnel plot should be used to assess publication bias. The certainty of synthesised outcome(s) should also be considered using a validated framework (e.g. GRADE).

7. The discussion fails to comment on the limitations of either the evidence or review process, emphasising the unknown exposure to risk of bias.

8. Finally, If revising the manuscript, clear statements in the manuscript aligning the study with the PRISMA criterion would be appreciated by readers (e.g., missing individual data through reporting).

Reviewer #2 (Remarks to the Author):

Of the many challenges associated with malaria, the assessment of disease severity is of particular importance. The microscopic quantification of Plasmodium parasites is often an unreliable marker of infection progression. The authors present a convincing case about the prognostic value of counting haemozoin-containing polymorphonuclear leukocytes and highlight the need of additional markers of disease severity. The study utilizes a substantial data set of 5 different trials with the inclusion of children and adults diagnosed with severe malaria in Asia or Africa. Their findings corroborate previous

reports about the prominence of misdiagnosed severe malaria among African children. The consequent mistreatment can result in higher mortality and misuse of antimalarials, risking further resistance of the malaria parasites. I believe the conclusions presented in this article are well-founded and significant, corroborated by comprehensive statistical analysis. Their proposed method of evaluation is fast, easy to perform and only requires materials already available in most affected regions.

Suggested improvements:

The article mentions that the use of rapid diagnostic tests is on the rise. I think the authors should consider including some information about the issues related to HRP2 deletions (an example: <https://elifesciences.org/articles/25008>) and how this affects the value of microscopy-based diagnosis.

I have the following minor comments related to figures:

- Table 1: Consider moving it to the Supplementary section.
- Figures 1-2: The PMN/PMM vs Mortality graphs have grid lines that do not match the ticks on the X axis. I suggest adjusting the grids to the ticks for improved clarity and because PMN levels above 5% have a lot of significance throughout the article, which would be better emphasized on the graph.
- Figure 3: I think this figure is a bit too data-heavy. Consider presenting it in a more compact way or moving it to the Supplementary section.
- Figure 5: Top right panel X axis misses tick '10'. Consider adding a gap between 1 and 3 on the X axis of the right panels for the better utilization of graph space.
- Figure S2: Consider moving it to the main section.
- Figure S4: The Y axis title on the bottom right panel has PMN instead of PMM.

Reviewer #3 (Remarks to the Author):

Review Srinamon et al "The prognostic and diagnostic value of intraleukocytic malaria pigment: an individual patient data pooled meta-analysis of 32,000 patients with severe malaria in Africa and Asia"

Srinamon et al have conducted a meta-analysis of the association of pigmented leukocytes with in-hospital mortality using data collected as part of clinical trials and observational studies across multiple countries in both Asia and Africa. They describe a positive association between pigmented polymorphonuclear cells and mortality and provide evidence that there is a prognostic value of pigmented leukocytes in addition to clinical and other laboratory markers. This paper would be of

interest to lab technicians and clinicians working in malaria endemic areas as well as researchers in the field of severe malaria and disease pathology.

Overall, the paper is well written and easy to follow. The rationale of the work is clear and convincing. Methods of data collection and analysis to my knowledge are appropriate and the results are well presented. The results are noteworthy and support the conclusions. The discussion does a good job at putting into context the collection and use of PMN data in a clinical situation, but it does not discuss the strengths and weaknesses of the study nor try to explain why results from this analysis vary from previously published results.

Specific comments

The authors use >5% pigmented PMN or monocytes as a cut off in a few of the analysis but why this cut off was used is unclear. Is this a cut off which has been used previously? Was it to do with distribution of the data?

Line 128: "thus, only a minority of the children in the SMAC study had strictly defines severe malaria" as the paragraph is currently worded, it is unclear to me why only a minority of the children had severe malaria. Please reword and clarify.

Lines 277-271: Please expand on the line "in African children the additional prognostic value was substantial" in the context of figure 4.

Figure 4A: please reorder the labels in the box so they are in the same order as in the graphs, when printed in black and white it is easy to misread.

Line 322: incomplete sentence, please reword

Line 350: please provide a reference for PMN counts during malaria

Line 374: please discuss why results from the previous SMAC analysis differed from the results presented in this manuscript

A number of points from the Prisma Checklist are yet to be included or could be better clarified

Item 2: Abstract- several details applicable to this study should be included if possible, including

Background- Outcome should be clearly stated.

Methods- data sources should be mentioned.

Results- numbers and types of studies and participant numbers should be given

Item 3: Clarify the outcome when stating the objective in the introduction.

Item 10: describe how data were requested, collected and managed (note: page 17 of the submitted manuscript which is given as a reference for this item is SFigure 2)

Item 15: Apologies- but could the authors please explain/justify why the risk of bias across studies is not applicable.

Item 24: Please specify the strength of evidence for the main outcomes

Item 25: Strengths and limitations are not discussed

Reviewer #4 (Remarks to the Author):

This is an individual patient meta-analysis about the diagnostic and prognostic value of pigment in white blood cells in/for patients with malaria. It is interesting to see how this study differs from previous studies and how the conclusions were drawn. However, I do not think the conclusions made in this study really hold. The provided data have more than enough power to show any effects, but the interpretation of these effects is a bit overoptimistic.

1. I found the title a bit misleading for two reasons: (1) the diagnostic value was only a small part of the study and (2) the term individual patient data meta-analysis implies a more rigorous systematic review process and implies more than a relatively random three to five studies included. Also, I am not sure if this study is really about the prognostic and diagnostic value of malaria pigment in cells: I think the prognostic element should be explained more clearly (see following comment); and I think the “diagnostic value” is more the association between infection load and pigmented cells. If I am correct, then I think the title should be rephrased.

2. The authors assess the prognostic value as the odds ratio between pigmented cells and mortality. But adjusted for a number of factors, without explaining why this adjustment is done and whether these factors are seen as effect modifiers, confounders or other sort of factors. It is not clear why this adjustment is done, and neither is it clear how the prognostic value in that case should be interpreted (more as a predictive value or as a verdict about causality?). Please clarify.

3. The authors assess the diagnostic value as the difference in mean PfHRP2 for each percentage of pigmented cells. But this is only an indication that there is an association and says nothing about the diagnostic value of the pigmented cells: it says nothing about whether pigmented cells can be used to distinguish patients with severe malaria from those without severe malaria. I suppose that the higher PfHRP2 concentrations correlate with more severe malaria, but I also suspect that this is not a linear not absolute relationship. Furthermore, using the term ‘diagnostic’ may seem a bit odd for detecting a state

of disease, rather than the disease itself. This should at least be acknowledged, but maybe the diagnostic value part is not really necessary.

4. Methods: IPD meta-analysis suggests that there is a systematic review underlying the selection of studies and patients. The authors conducted indeed a systematic review, but using a very narrow search strategy and no methodological quality assessment of the included studies. Please provide the full search strategy, included the search dates, and preferably multiple bibliographic databases. Also, I think the search terms should be broadened, so that maybe more studies will be retrieved.

5. Methods: The authors dichotomized the pigmented cells percentage. This means that quite some information gets lost. Would it be possible to keep the pigmented cell percentages as a continuous variable?

6. Methods: why do the authors do an IPD and a meta-analysis?

7. Methods: is it possible to provide additional information about the analyses? In any case about the diagnostic value, as this part is now missing from the methods.

8. There seems to be quite some variation in how the different variables were measured and/or defined. This may also have led to misclassification (as the authors also suggest). How may this have impacted the results and could the authors mention this as a limitation of their study?

9. The result of the meta-analysis is an OR of 2.67. Would it perhaps be possible to provide an interpretation of this number in clinical terms? It means that the odds of dying is 2.67 times as large for those with more pigmented cells than those with fewer pigmented cells, but what should be the consequences of this? It may not be sufficient to use the percentage pigmented cells as an indicator to tell the family that the patient is probably going to die... So what is the relevance of this finding?

10. I do not agree with the conclusion: "The proportion of PMNs containing malaria pigment has strong diagnostic and prognostic value, and importantly it adds significantly to the bedside assessment of coma and acidotic breathing." I think an OR of 2.67 is not "strong" and also the association between pigmented cells and PfHRP2 is not necessarily "strong". Please provide a better argumentation of why this is a strong value, or moderate the conclusion.

Peer review done by Mariska Leeflang

Reviewer #1 (Remarks to the Author):

The paper seeks to show that the proportion of pigment containing polymorphonuclear leukocytes in peripheral blood films is strongly positively correlated with prognosis in malaria. Whilst for those of us working in malaria diagnostics the motivation for the study is clear, there is currently a lack of clarity over what is being claimed, with conclusions drawn often being qualified.

It is probable that the paper would be of interest to those with a specialist knowledge of malaria testing. As written, the messages within the text do not provide clear unambiguous results that would be of interest to a broader readership associated with this journal. It is an output that may sit more naturally in a specialist journal such as *Malaria*, where the nuances of the messaging might be better understood.

We have rewritten the manuscript to improve clarity but, with respect, disagree that the findings of this study are of limited relevance. Severe malaria is the cause of nearly 2000 deaths each day, mainly in African children. Surely a simple, affordable, available, rapid, readily performed, independent method of diagnosis and prognostic assessment of one of the main causes of preventable childhood death in tropical countries is more than of peripheral interest?

There are a number of technical clarifications that should be addressed by the authors.

1. The study selection process is, as written, unclear. The sourcing of the seemingly pre-defined AQ Vietnam, SEAQUAMAT and AQUAMAT, in relation to the systematic review studies (SMAC, Lyke et al. and Boeuf et al.) should be clarified. This raises a concern around the comprehensive sampling of the evidence-base. The authors should consider replicating their search on other databases (at least one other of MEDLINE/ EMBASE/ etc.).

We agree that the study selection was not clear in the original submission. There are two parts to this manuscript. In the first part, we provide new data on intraleukocytic pigment counts from three studies run/coordinated by the Mahidol Oxford Tropical Medicine Research Unit in Thailand and the Oxford University Clinical Research Unit in Vietnam (for the AQ Vietnam trial, only data from the first 300 patients were published previously; the pigment count data from the SEAQUAMAT and AQUAMAT trials have never been published before). We designed and coordinated these three studies which represent the main evidence base for the current treatment of severe malaria. Slides had been kept from these studies and so we could assess intraleukocytic pigment in the majority of patients (some slides were lost, some were unreadable).

In the second part of the report, we did a systematic review to identify all studies that have measured intraleukocytic pigment in patients with severe malaria, and then added the data from the studies with over 100 patients to our analyses. It is important to emphasise that this comprises the majority of all prospectively studied patients with severe malaria reported in the medical literature in the past 50 years.

In light of the reviewer's comment, we did the following to ensure that all relevant studies were identified:

- We replicated the search on EMBASE;
- We reviewed all studies included in a recent systematic review of prognostic factors in African children with severe malaria (Sypniewska *et al* BMC Med 2017);
- We searched Google Scholar for papers that had cited the original manuscript on the prognostic value of intraleukocytic pigment (Phu *et al* 1995).
- Following a comment from reviewer 4 we added the additional term "haemozoin" to the literature searches (this helped identify a few extra relevant studies).

These changes resulted in 68 additional papers identified (a total of 202), of which 19 had recorded intraleukocytic pigment counts in patients diagnosed with severe malaria. All the additional reports were identified by the extra search term “haemozoin”. Only 2 of these additional studies had over 100 patients (Luty *et al* [2000], n=100; Birhanu *et al* [2017], n=102). However, the corresponding authors for Luty *et al* could not find the raw data so this study’s data could not be included. The contact details for the corresponding author for Birhanu *et al* were no longer functional and so we failed to get a reply to our request for data.

Thus, the underlying database for the meta-analysis has not changed since the original submission. We are very confident that we have done an exhaustive search of the medical literature and that we have not missed any major studies with more than 100 patients which recorded intraleukocytic pigment.

2. The methodology of the literature screening is not clearly reported - this should ideally be replicated by 1 or more authors and the process should be summarised with a PRISMA flow diagram, as well as in the Nature Communication study protocol.

A second reviewer (KS) replicated the screening of all abstracts. A third reviewer resolved disagreements (NJW). We have added a flow diagram to the supplementary materials (Fig S3).

3. A concern arises around the meta-analysis populations. Firstly, although inclusion and exclusion criteria are stated clearly in the review, importantly, SMAC is a retrospective analysis. This may be at odds with the inclusion criteria sought, as registered on Prospero, thus challenging the conduct of the review. Differences between studies also introduce concerns around the internal validity of the analysis. Inclusion and exclusion criteria for each study are not clearly stated, nor is how the analysis was adapted for missing patient data (especially PMN/ PMM counts).

In the Discussion of the original submission we had written (regarding SMAC):

“Their retrospective data analysis concluded that “although high levels of pigmented cells were associated with a fatal outcome in some study sites, they were not useful predictors of outcome across Africa” and that “they were not useful markers of fatal outcome for individual patients”.

The word “retrospective” here was misplaced. Determining the prognostic value of intraleukocytic pigment was in fact the primary aim of the SMAC study and SMAC was not a retrospective cohort study (patients were prospectively enrolled). SMAC certainly fits the criteria of our Prospero plan. We have therefore removed the word “retrospective”

The inclusion and exclusion criteria have been updated for all three randomised trials.

We have added to the Methods that only patients with readable and available slides (who could have pigment containing PMN counts quantified) were included in the analysis.

4. Focussing specifically on the SMAC analysis, this study is both significantly larger (n=26,389) and presents a patient population with less severe malaria (4% mortality, 9% coma). Although in the discussion the authors state that they conducted their analysis on a severe malaria subgroup, the criteria and baseline characteristics for this group are not clarified in the text. Given that this is used to evidence the validity of the prognostic indication of >5% pPMN, it may raise concerns around selection bias in this cohort. Additionally, the smaller subgroup may indicate a smaller analysed population than the title indicates.

We apologise for lack of clarity in the discussion and have therefore rewritten this description. We did not analyse a subgroup of SMAC, we analysed all patients. The discussion now reads:

Our conclusions differ from those of the SMAC network analysis [12]. Their data analysis concluded that “although high levels of pigmented cells were associated with a fatal outcome in some study sites, they were not useful predictors of outcome across Africa” and that “they were not useful markers of fatal outcome for individual patients”. Although the children studied in SMAC had a much broader range of disease severities (i.e. in contrast to the randomised controlled trials evaluated here, a substantial proportion of children in the SMAC database likely did not have severe malaria and the overall mortality was low), and thick blood films rather than thin films were evaluated, reanalysis of the SMAC data shows clear prognostic value intraleukocytic pigment, consistent with the four other malaria studies (Supplementary Fig. S8).

Although SMAC was fairly different from the other studies (lower severity and cell counts done on thick films) the results are very similar. We have added a forest plot to the Supplementary Materials (Fig. S8) showing only the data from the SMAC study.

5. A further concern arises in the geographic heterogeneity between Asian and African patient populations and whether this confounds the meta-analysis conducted. There may be a risk of substantial heterogeneity in Figure 3 ($I^2=68\%$), which further evidences this; although, this is not commented upon by the authors nor does there seem to be any methodological contingencies to explore this: pre-specified or ad-hoc. Similarly, there is clear inconsistency in the age of patients between studies. This is important as immunological maturation may influence malarial disease course and sequelae, possibly further confounding the internal validity of this analysis.

We agree with the reviewer that this is an important point which should have been given more emphasis in the manuscript. There clearly is a substantial amount of heterogeneity across studies. However, the random effects model and the common effects model give approximately the same result (odds ratio for death of ~ 2.5 in those with $>5\%$ pigment containing PMNs). Not a single study (or study site in the large multicentre trials) excludes odds ratios of this magnitude or greater. There is no mechanistic reason why the prognostic value of pigment containing PMNs should differ between adults and children, apart from the differences in the specificity of the diagnosis (related to transmission intensity, for which patient age is a proxy). Nearly all the children included in the meta-analysis are from sub-Saharan Africa (Lyke et al, SMAC and AQUAMAT), whereas all the adults are from the two trials in Asia (AQ Vietnam and SEAQUAMAT). As is clear from Fig. 1, the prognostic utility of intraleukocytic pigment is stronger in adults. This is most likely to be due to differences in diagnostic accuracy: the diagnosis of severe malaria in the studies in Asia is very specific, whereas in high transmission areas in Africa it is much less so. If we look separately at children (age <15 years) versus adults (age >15 years), the odds ratio for death is approximately 2.5 in children (under a random effects model; 2.2 under a fixed effects model) versus 3.5 in adults (under a random effects model; 3.1 under a fixed effects model). This difference is significant when coded as an interaction in a logistic regression model.

We have added these subgroup forest plots to the Supplementary Materials along with the following sentence in the Results (lines 135-140):

“There was heterogeneity in the prognostic value of $>5\%$ pigment containing PMNs when analysing adults versus children (>15 years vs <15 years). As apparent from Fig. 1, the prognostic value was greater in Asia (predominantly adults) as compared to Africa (all children). The odds-ratio for death in the pooled data set adults was 3.44 for adults (95% CI: 2.08-5.69, Supplementary Fig. S5) and 2.37

for children (95% CI: 2.37-5.99, Supplementary Fig. S6), a significant difference under an interaction model ($p=0.002$)."

6. Noticeably, there appears to be no risk of bias assessments (or references to such) at either the study or outcome levels of the review. This may indicate that the authors have not critically considered sources of variation or error in their prognostic modelling and challenges the validity of the work they have undertaken. For each study included, a risk of bias evaluation (ideally with an externally validated tool) should have been conducted for each outcome extracted and, additionally, where synthesised, a funnel plot should be used to assess publication bias. The certainty of synthesised outcome(s) should also be considered using a validated framework (e.g. GRADE).

We have added a risk of bias assessment using an adapted version of the QUADAS-2 tool (see supplementary materials)

7. The discussion fails to comment on the limitations of either the evidence or review process, emphasising the unknown exposure to risk of bias.

We have added a paragraph in the Discussion on the study limitations. The main limitation (poor diagnostic accuracy) is also highlighted in the quality assessment in the supplementary materials.

8. Finally, if revising the manuscript, clear statements in the manuscript aligning the study with the PRISMA criterion would be appreciated by readers (e.g., missing individual data through reporting).

We have updated the PRISMA checklist and added the missing elements.

Reviewer #2 (Remarks to the Author):

Of the many challenges associated with malaria, the assessment of disease severity is of particular importance. The microscopic quantification of Plasmodium parasites is often an unreliable marker of infection progression. The authors present a convincing case about the prognostic value of counting haemozoin-containing polymorphonuclear leukocytes and highlight the need of additional markers of disease severity. The study utilizes a substantial data set of 5 different trials with the inclusion of children and adults diagnosed with severe malaria in Asia or Africa. Their findings corroborate previous reports about the prominence of misdiagnosed severe malaria among African children. The consequent mistreatment can result in higher mortality and misuse of antimalarials, risking further resistance of the malaria parasites. I believe the conclusions presented in this article are well-founded and significant, corroborated by comprehensive statistical analysis. Their proposed method of evaluation is fast, easy to perform and only requires materials already available in most affected regions.

We thank the reviewer for their kind comments.

Suggested improvements:

The article mentions that the use of rapid diagnostic tests is on the rise. I think the authors should consider including some information about the issues related to HRP2 deletions (an example: <https://elifesciences.org/articles/25008>) and how this affects the value of microscopy-based diagnosis.

We have now added this to the Discussion (lines 192-194).

I have the following minor comments related to figures:

- Table 1: Consider moving it to the Supplementary section.

This has been moved as suggested.

- Figures 1-2: The PMN/PMM vs Mortality graphs have grid lines that do not match the ticks on the X axis. I suggest adjusting the grids to the ticks for improved clarity and because PMN levels above 5% have a lot of significance throughout the article, which would be better emphasized on the graph.

We thank the reviewer for this suggestion. Figs 1-2 have been amended so that the ticks are on the log scale, highlighting the 5% threshold.

- Figure 3: I think this figure is a bit too data-heavy. Consider presenting it in a more compact way or moving it to the Supplementary section.

We agree that there was too much information in this Figure. We have removed two of right columns (CI and weight in the random effects model).

- Figure 5: Top right panel X axis misses tick '10'. Consider adding a gap between 1 and 3 on the X axis of the right panels for the better utilization of graph space.

We have amended this Figure accordingly.

- Figure S2: Consider moving it to the main section.

This has been moved.

- Figure S4: The Y axis title on the bottom right panel has PMN instead of PMM.

Thank you for spotting this error! To avoid confusion between the two acronyms we have changed “PMM” to “monocytes” throughout the manuscript.

Reviewer #3 (Remarks to the Author):

Review Srinamon et al "The prognostic and diagnostic value of intraleukocytic malaria pigment: an individual patient data pooled meta-analysis of 32,000 patients with severe malaria in Africa and Asia"

Srinamon et al have conducted a meta-analysis of the association of pigmented leukocytes with in-hospital mortality using data collected as part of clinical trials and observational studies across multiple countries in both Asia and Africa. They describe a positive association between pigmented polymorphonuclear cells and mortality and provide evidence that there is a prognostic value of pigmented leukocytes in addition to clinical and other laboratory markers. This paper would be of interest to lab technicians and clinicians working in malaria endemic areas as well as researchers in the field of severe malaria and disease pathology.

Overall, the paper is well written and easy to follow. The rationale of the work is clear and convincing. Methods of data collection and analysis to my knowledge are appropriate and the results are well presented. The results are noteworthy and support the conclusions. The discussion does a good job at putting into context the collection and use of PMN data in a clinical situation, but it does not discuss the strengths and weaknesses of the study nor try to explain why results from this analysis vary from previously published results.

We thank the reviewer for their kind comments. We have expanded on the limitations (penultimate paragraph of the Discussion). We have now added brief summaries of all relevant previous series in the Supplementary Materials. Interestingly there is relatively little variation in results, it is the interpretations which differ. Most previous studies focussed on pigment containing leukocytes as a marker for severe anaemia or other manifestations of severe malaria. The notable outlier in terms of prognostic value is the report of the SMAC study by Kremsner *et al* which concluded that pigment containing neutrophil counts did not have prognostic value above that provided by a simple bedside assessment. This was a mistake. Reanalysis of these data shows unequivocally the independent prognostic value of pigment containing neutrophil counts and is entirely consistent with the new studies reported here (see Supplementary Fig S8). This is now explained in more detail.

Specific comments

The authors use >5% pigmented PMN or monocytes as a cut off in a few of the analysis but why this cut off was used is unclear. Is this a cut off which has been used previously? Was it to do with distribution of the data?

We agree that this is an important point which we had not explained. In the first report of pigment containing PMNs (Phu et al 1995), 5% was proposed as a cut-off for clinical practice. We have added to the Methods (lines 437-440):

"The primary exposure definition was >5% pigment containing PMNs or PMMs. The 5% threshold was proposed in the first paper noting the prognostic value of PMN and PMM counts [8]. In addition, it is easy to remember and thus appropriate for use in clinical practice."

We have added to the Results (lines 87-91):

"In the initial report demonstrating the prognostic value of pigment containing PMN counts a 5% threshold was proposed for clinical practice [8]. We re-evaluated the prognostic value of this threshold in this larger patient population. Under a mixed effects logistic regression model, combining individual patient data from the three randomised trials (n=6027), greater than 5% pigment containing PMNs was associated with an odds-ratio for death of 2.39 (95% CI 2.03-2.82)."

Line 128: “thus, only a minority of the children in the SMAC study had strictly defined severe malaria” as the paragraph is currently worded, it is unclear to me why only a minority of the children had severe malaria. Please reword and clarify.

The word “thus” was misplaced. We have rephrased this as:

*“All children aged between 1 month and 15 years of age suspected of having malaria and who were sick enough to be hospitalized were screened with a thick blood smear for *P. falciparum* parasites. Only a minority of the children in the SMAC study had strictly defined severe malaria [1].”*

This is because SMAC was an inclusive study based initially on physician assessment. Hospitalisation + parasites on a thick film is not sufficient for a strict severe malaria definition (WHO guidelines 2014).

Lines 277-271: Please expand on the line “in African children the additional prognostic value was substantial” in the context of figure 4.

We have removed this sentence as it was not justified.

Figure 4A: please reorder the labels in the box so they are in the same order as in the graphs, when printed in black and white it is easy to misread.

We thank the reviewer for pointing this out – this has been changed.

Line 322: incomplete sentence, please reword

We have re-written the Discussion substantially – we hope this is clearer now.

Line 350: please provide a reference for PMN counts during malaria

We have added a reference (Warrell et al NEJM 1982).

Line 374: please discuss why results from the previous SMAC analysis differed from the results presented in this manuscript

As explained above this was a mistake. Re-analysis of the individual patient SMAC data shows unequivocally the independent prognostic value of pigment containing neutrophil counts and is entirely consistent with the new studies reported here. We have added a supplementary Figure showing a forest plot for the SMAC study on its own (Fig. S8).

A number of points from the Prisma Checklist are yet to be included or could be better clarified.

We have added the missing components to the PRISMA checklist.

Item 2: Abstract- several details applicable to this study should be included if possible, including:

Background- Outcome should be clearly stated.

Methods- data sources should be mentioned.

Results- numbers and types of studies and participant numbers should be given

We have re-written the abstract but we are constrained to the Nature Communications abstract requirements (150 words, non-structured).

Item 3: Clarify the outcome when stating the objective in the introduction.

In the last paragraph of the introduction we have changed the sentence to:

“Following a systematic review of the literature, we pooled individual patient data from over 32,000 patients clinically diagnosed with severe falciparum malaria and assessed the prognostic value for in-hospital mortality of the proportion of pigment containing PMNs and PMMs.”

Item 10: describe how data were requested, collected and managed (note: page 17 of the submitted manuscript which is given as a reference for this item is SFigure 2)

We have amended this item.

Item 15: Apologies- but could the authors please explain/justify why the risk of bias across studies is not applicable.

We have now added the QUADAS assessment

Item 24: Please specify the strength of evidence for the main outcomes

We have amended this item.

Item 25: Strengths and limitations are not discussed

We have re-emphasised the main strength of the paper which is that it includes the majority of all patients with severe malaria who have been studied prospectively in the past 50 years. This limits concerns about publication bias and strengthens generalisability. As described above limitations have been expanded.

Reviewer #4 (Remarks to the Author):

This is an individual patient meta-analysis about the diagnostic and prognostic value of pigment in white blood cells in/for patients with malaria. It is interesting to see how this study differs from previous studies and how the conclusions were drawn. However, I do not think the conclusions made in this study really hold.

We have tried to clarify that this is the largest series ever reported in severe malaria and the results are highly consistent across studies. Having >5% pigment containing neutrophils does not mean the patient will die: it means that the diagnosis of severe malaria is likely to be correct and that they are at a high risk of death and need rapid administration of parenteral artesunate.

The provided data have more than enough power to show any effects, but the interpretation of these effects is a bit overoptimistic.

1. I found the title a bit misleading for two reasons: (1) the diagnostic value was only a small part of the study and (2) the term individual patient data meta-analysis implies a more rigorous systematic review process and implies more than a relatively random three to five studies included.

The three studies with new data are not “relatively random”. These three RCTs are the three largest randomised controlled trials in the antimalarial treatment of severe malaria ever conducted. They represent the main evidence base behind the current treatment of severe malaria. SMAC is the largest ever observational study of severe malaria. Together the pooled data set represents the majority of all prospectively studied patients with severe malaria in the last half century.

Also, I am not sure if this study is really about the prognostic and diagnostic value of malaria pigment in cells: I think the prognostic element should be explained more clearly (see following comment); and I think the “diagnostic value” is more the association between infection load and pigmented cells. If I am correct, then I think the title should be rephrased.

Thank you for these comments. We have tried to rebalance the presentation to clarify the message (which is a simple one). It is true that the diagnostic value is only one paragraph in the Results. But it is an extremely important paragraph. A substantial number of African children are misdiagnosed with severe malaria, resulting in preventable death (White et al, Lancet 2022). Current estimates are that one third of children studied in specialist research centres are misdiagnosed. This is a huge number accounting for approximately 200,000 deaths per year. It is likely that many or most have sepsis (see ref 23: Gilchrist et al). This has huge clinical and epidemiological significance. Finding intraleukocytic pigment helps differentiate between true severe malaria and sepsis. Surely this is important?

In previous work (Hendriksen et al 2012, Watson et al 2022) we have shown that plasma *Pf*HRP2 is the best available diagnostic marker for severe malaria. However, measuring plasma *Pf*HRP2 in remote settings is not yet possible. The AQUAMAT trial is the largest series of patients with plasma *Pf*HRP2 measured (over 3,600 patients – this was a huge endeavour). We show that intraleukocytic pigment (much easier and cheaper to measure) is highly predictive of a high plasma *Pf*HRP2 (>1,000 ng/mL). This provides additional evidence that intraleukocytic pigment is a useful proxy measurement of the infection load. The total parasite biomass is the primary determinant of severity.

2. The authors assess the prognostic value as the odds ratio between pigmented cells and mortality. But adjusted for a number of factors, without explaining why this adjustment is done and whether

these factors are seen as effect modifiers, confounders or other sort of factors. It is not clear why this adjustment is done, and neither is it clear how the prognostic value in that case should be interpreted (more as a predictive value or as a verdict about causality?). Please clarify.

We thank the reviewer for this important comment. We agree that it is important to lay down our *a priori* causal structure. We have added a Supplementary Figure (Fig. S9) which shows a proposed causal diagram. Under this causal model, the clinical variables (coma, acidosis) mediate the relationship between the parasite biomass (the major determinant of severity in severe malaria) and death. The adjusted analysis thus allows us to estimate direct effects between the proxy measurements for parasite biomass (intra-leukocytic pigment) and death (the mediators are imperfectly measured).

3. The authors assess the diagnostic value as the difference in mean PfHRP2 for each percentage of pigmented cells. But this is only an indication that there is an association and says nothing about the diagnostic value of the pigmented cells: it says nothing about whether pigmented cells can be used to distinguish patients with severe malaria from those without severe malaria. I suppose that the higher PfHRP2 concentrations correlate with more severe malaria, but I also suspect that this is not a linear not absolute relationship. Furthermore, using the term 'diagnostic' may seem a bit odd for detecting a state of disease, rather than the disease itself. This should at least be acknowledged, but maybe the diagnostic value part is not really necessary.

As described above higher proportions of pigment containing leukocytes are associated specifically with "true" severe malaria and differentiate this from sepsis or other causes of severe illness and incidental parasitaemia. This is a very important practical point.

4. Methods: IPD meta-analysis suggests that there is a systematic review underlying the selection of studies and patients. The authors conducted indeed a systematic review, but using a very narrow search strategy and no methodological quality assessment of the included studies. Please provide the full search strategy, included the search dates, and preferably multiple bibliographic databases. Also, I think the search terms should be broadened, so that maybe more studies will be retrieved.

This is now included. We added "haemozoin" to the search which retrieved 2 additional relevant studies (unfortunately for neither could the data be obtained).

5. Methods: The authors dichotomized the pigmented cells percentage. This means that quite some information gets lost. Would it be possible to keep the pigmented cell percentages as a continuous variable?

Figures 1,2, and 6 show non-dichotomised relationships. Figure 3 shows odds-ratios for 4 bins relative to a zero count. We agree that the continuous data are "better", however in clinical practice it is necessary to use simple threshold values (these are easy to remember). For this reason we focus on the 5% threshold which was pre-specified (Figure 4).

6. Methods: why do the authors do an IPD and a meta-analysis?

The individual patient data allows us to look at the relationship between the continuous count data (Figures 1,2 and 6) and the outcome. It also allows us to adjust for individual patient baseline variables (Figure 5). As noted above, estimating the simple odds-ratio for death using a 5% threshold is important for implementation.

7. Methods: is it possible to provide additional information about the analyses? In any case about the diagnostic value, as this part is now missing from the methods.

Thank for this comment. We have added a section in the Methods about the diagnostic analysis.

8. There seems to be quite some variation in how the different variables were measured and/or defined. This may also have led to misclassification (as the authors also suggest). How may this have impacted the results and could the authors mention this as a limitation of their study?

We have added this as a limitation in the Discussion.

9. The result of the meta-analysis is an OR of 2.67. Would it perhaps be possible to provide an interpretation of this number in clinical terms? It means that the odds of dying is 2.67 times as large for those with more pigmented cells than those with fewer pigmented cells, but what should be the consequences of this? It may not be sufficient to use the percentage pigmented cells as an indicator to tell the family that the patient is probably going to die... So what is the relevance of this finding?

In the busy health centres and hospitals which admit severely ill children in tropical countries, there are insufficient nurses, doctors and laboratory technicians. There are also usually limited resources in terms of drugs, equipment, oxygen, specialist beds etc. Only rapidly performed simple blood tests are useful in informing clinical management and triage. The first question is "what is the cause of fever". Many children have incidental low parasitaemias and so may be RDT or slide positive, but do they have severe malaria? This critical question is informed by the count of pigment containing leukocytes (particularly neutrophils). Second, not all children can be admitted to the high dependency unit or beds, or the intensive care unit. This is the value of a simple prognosticator – in focussing the limited resources on those in greatest need.

10. I do not agree with the conclusion: "The proportion of PMNs containing malaria pigment has strong diagnostic and prognostic value, and importantly it adds significantly to the bedside assessment of coma and acidotic breathing.". I think an OR of 2.67 is not "strong" and also the association between pigmented cells and PfHRP2 is not necessarily "strong". Please provide a better argumentation of why this is a strong value, or moderate the conclusion.

We respectfully disagree. We have explained the clinical context above. A simple, rapid, available and affordable blood test provides, within minutes, information which guides the clinician in the choice of treatment, supportive management and triage. This odds ratio is comparable to the other established prognosticators.

REVIEWER COMMENTS

Reviewer #1 (Remarks to the Author):

The authors have given detailed responses, although I remained concerned that the messaging in this paper being nuanced.

For example, in response to the point on geographical heterogeneity and to those of Referee 4 (point 10) there are still open questions that the OR of 2.67 was not "strong." The authors argue that the OR "is comparable to the other established prognosticators". If this is the case, can the authors articulate what the step-change is?

In the context of geographical heterogeneity the authors provide more detail giving analysis for pooled populations that indicates that for children (where diagnosis is particular important) the OR was 2.37 – less than the case for adults or existing methods.

Reviewer #4 (Remarks to the Author):

There is no doubt that prognosis of mortality risk in severe malaria and the differentiation between malaria and sepsis is extremely important and that clinicians, technicians and patients are waiting for good prognostic and diagnostic markers. However, about what a good marker entails, opinions may differ. The authors of this manuscript have addressed few of my previous comments and I would like to explain why I have still doubts about the value of the here presented markers.

1. If I understand correctly, then the adjustments in the prognostic model do not account for confounding (as pigmentation is not seen as the "exposure", according to the causal diagram), but are a way to express the OR of pigmentation on top of variables like coma and acidosis. Which is fine. The authors conclude that "The results show clearly that pigment containing PMN counts provided additional prognostic information over that of the bedside clinical assessment and even laboratory biomarkers", based on Figure 5. But Figure 5 is not that convincing: as soon as the figures are adjusted, then the ORs decline and comes closer to 1.

2. The authors also state that: "Assuming a baseline risk of death of 5%, an odds-ratio for death of 2.6 translates to a risk ratio of approximately 2.5, i.e. a mortality of 12.5% in patients with >5% pigment

containing PMNs compared to a mortality of 5% in patients with $\leq 5\%$ pigment containing PMNs". I think these numbers are very helpful, thank you for the explanation. But how does that translate to practice, if the doctor knows the patient may have a risk of dying of 12.5% instead of 5%? Apologies for my ignorance, but although I can imagine that this is extremely important information, I would think that the care for both groups would really change because of these numbers. And I also do not see how knowing this, would prevent people with severe malaria from dying. I mean, if I am caring for 500 patients, of whom 100 have hyperpigmented PMNs (mortality risk 12.5% and thus 13 will die) and 400 have not (mortality risk 12.5% and thus 20 will die), I am not sure if I would treat those without hyperpigmented PMNs differently than those with. And looking at the studies included in the meta-analyses, the balance may be even more tilted than in my hypothetical example.

3. A similar remark I have about the diagnostic value of pigmented PMNs. The authors state that a higher percentage of patients with hyperpigmented PMNs had high parasitic loads (583 out of 755) than of patients with lower pigmentation rates (1306 out of 2178). Similarly with my previous comment, if we were to use pigmented PMNs as diagnostic marker and treat those with high pigmentation rates differently from those without, then we may do more people harm ($1306/2933=45\%$) than good ($583/2933=20\%$).

4. Another way to look at the diagnostic value of the markers, is through the diagnostic accuracy paradigm (for which the authors used QUADAS-2, after all). With the numbers presented in the main text, we would have 583 true positives (high rate pigmented PMN and high parasitic load), 172 false positives (high rate+low parasites), 1306 false negatives (low rates + high parasitic load), 872 true negatives. That would lead to a sensitivity of $583/(583+1306)=31\%$ and a specificity of $872/(872+172)=84\%$. Which to me does not sound as a very promising diagnostic marker.

Again, I am not saying that severe malaria is not important; I am also not saying that pigmentation of PMNs or monocytes is useless. But to state that they have strong prognostic and diagnostic value would be overstating the findings of this study, in my humble opinion.

Thank you for giving us the opportunity to respond to the reviewers' comments. The main concern is that we overstated our case, and reviewer 4 feels that the clinical relevance of the independent diagnostic and prognostic measure is uncertain. We have attenuated the description of the strength of the result throughout, removing the word "strongly" and simply presenting the findings. In order to explain the clinical relevance of our findings we have rewritten the first half of the discussion to describe the utility of this measure in its clinical context. We hope these changes will now make the paper suitable for publication. Our detailed responses are appended below.

REVIEWER COMMENTS

Reviewer #1 (Remarks to the Author):

The authors have given detailed responses, although I remained concerned that the messaging in this paper being nuanced.

For example, in response to the point on geographical heterogeneity and to those of Referee 4 (point 10) there are still open questions that the OR of 2.67 was not "strong." The authors argue that the OR "is comparable to the other established prognosticators". If this is the case, can the authors articulate what the step-change is?

We are unsure what the reviewer means by step change? However, we have added a comparison with other established prognosticators (now given in the Supplementary Materials). The main established prognostic variables for severe malaria are: coma, acidosis, severe anaemia, and hyperparasitaemia. In our pooled dataset of African children (which is the largest ever reported):

1. Coma: odds ratio for death is 7.6 (19% vs 3% mortality in 31,000 children)
2. Respiratory distress (Kussmaul's breathing): odds ratio for death is 6.5 (19% vs 3.4% mortality in 31,400 children)
3. Laboratory defined acidosis (defined as either plasma lactate >5mmol or BUN >10mg/dL): odds ratio for death is 4.9 (11% vs 2.5% mortality in 29,000 children)
4. Severe anaemia (defined as haemoglobin < 5g/dL): odds ratio for death is 1.8 (7.7% vs 4.5% mortality in 30,600 children)
5. Hyperparasitaemia (defined as >100,000 parasites per uL): odds ratio for death is 1 (5.2% versus 5.2% mortality in 30,800 children)

These latter two are particularly important as they are the most common presentations in areas of high stable transmission where most deaths occur. From this perspective, having >5% pigment containing PMNs is a lesser prognosticator than acidosis and a better prognosticator than severe anaemia (a more common presentation in high transmission settings). However, this ignores important secondary information:

1. Coma, acidosis and severe anaemia have little diagnostic value in themselves without accompanying blood slide or rapid diagnostic test information;

2. There is a non-linear relationship between parasitaemia and death (very low and very high parasite densities are associated with high mortality because of the high rate of mis-diagnosis of severe malaria);
3. A high parasite count has some diagnostic value (ie it makes it more likely that malaria is the primary cause of illness) – this is shown now in Figure 6 (we have added parasite count to this Figure);
4. A high pigment containing PMN count has much greater diagnostic value (as expected from our mechanistic understanding of severe malaria, and from the strong association shown with plasma PfHRP2);
5. It is simple and very quick to count 100 PMNs on a thin blood film.

As we describe below, and have added extensively to the discussion, although this simple measure has independent prognostic value, its practical value is in the rapid assessment of a sick patient. In this emergency context (as discussed by the other reviewer) the value of this measure is not in isolation, but together with other indicators where it provides additional information above that provided by other measures. In practice, in the time-critical emergency assessment of severe malaria, the value of this test is not only that it has independent prognostic value, but that it is very simple, requires no additional procedures or equipment, takes less than a minute to obtain, and is therefore substantially quicker than all other laboratory tests. Delays are critical in the management of this medical emergency. The majority of deaths in children occur within 24 hours.

We have added the following sentence to the main text (lines 133-136):

“In African children, this was less than coma (odds-ratio of 7.6) and acidosis (Kussmaul's breathing: odds-ratio of 6.5), but greater than severe anaemia (odds-ratio of 1.8) and hyperparasitaemia (odds-ratio of 1), see Supplementary Materials.”

In the context of geographical heterogeneity the authors provide more detail giving analysis for pooled populations that indicates that for children (where diagnosis is particular important) the OR was 2.37 – less than the case for adults or existing methods.

Reviewer #4 (Remarks to the Author):

There is no doubt that prognosis of mortality risk in severe malaria and the differentiation between malaria and sepsis is extremely important and that clinicians, technicians and patients are waiting for good prognostic and diagnostic markers. However, about what a good marker entails, opinions may differ. The authors of this manuscript have addressed few of my previous comments and I would like to explain why I have still doubts about the value of the here presented markers.

We thank the reviewer for their comments. We agree that it is important to present the results accurately, without overselling them. There are two issues:

First is the issue of dichotomisation of a quantitative variable: as mentioned in the previous review comments, considering only whether pigment containing PMNs is greater or less than 5% loses much of the information. Our Figure 2 in the main text shows that mortality rises approx. log-linearly with the

percentage of pigment containing PMNs. Thus seeing 20% of pigment containing PMNs is very different from seeing 5%. As such it is different from inherently binary or categorical variables such as coma (or coma score) or respiratory distress. However, easy to remember threshold values are very useful in clinical practice, which is why we prespecified the analysis to focus on the 5% threshold.

Second, is the complex relationship between diagnostic value and prognostic value: in low transmission settings (most of Asia) the clinical diagnosis of severe malaria is highly specific, thus the proportion of pigment containing PMNs has high prognostic value (separating the different levels of severity of severe malaria) but presumably low diagnostic value (as there are very few false positives). However, in a high transmission setting (i.e. sub-Saharan Africa), clinical diagnosis has poor specificity, and in this setting diagnostic markers such as PfHRP2 or the % of pigment containing PMNs are very useful to separate out patients with true severe malaria and those with some other cause of severe illness. The misdiagnosed patients can also have a high mortality (severe sepsis has a worse prognosis than severe malaria) so the prognostic value is less.

1. If I understand correctly, then the adjustments in the prognostic model do not account for confounding (as pigmentation is not seen as the "exposure", according to the causal diagram), but are a way to express the OR of pigmentation on top of variables like coma and acidosis. Which is fine. The authors conclude that "The results show clearly that pigment containing PMN counts provided additional prognostic information over that of the bedside clinical assessment and even laboratory biomarkers", based on Figure 5. But Figure 5 is not that convincing: as soon as the figures are adjusted, then the ORs decline and comes closer to 1.

Yes – the variables are correlated so the effect sizes in the multivariable model decrease. The value in this measure is explained above -it provides additional information, and is very rapid to obtain. In real time it is therefore clinically very useful.

2. The authors also state that: "Assuming a baseline risk of death of 5%, an odds-ratio for death of 2.6 translates to a risk ratio of approximately 2.5, i.e. a mortality of 12.5% in patients with >5% pigment containing PMNs compared to a mortality of 5% in patients with ≤5% pigment containing PMNs". I think these numbers are very helpful, thank you for the explanation. But how does that translate to practice, if the doctor knows the patient may have a risk of dying of 12.5% instead of 5%? Apologies for my ignorance, but although I can imagine that this is extremely important information, I would think that the care for both groups would really change because of these numbers. And I also do not see how knowing this, would prevent people with severe malaria from dying. I mean, if I am caring for 500 patients, of whom 100 have hyperpigmented PMNs (mortality risk 12.5% and thus 13 will die) and 400 have not (mortality risk 5% and thus 20 will die), I am not sure if I would treat those without hyperpigmented PMNs differently that those with. And looking at the studies included in the meta-analyses, the balance may be even more tilted than in my hypothetical example.

Thank you for this important practical question which focusses on clinical management practices in busy under resourced hospitals. We accept that in a resource rich setting such as Europe or North America (where malaria admissions are unusual), there is a low threshold for admitting malaria patients to ICUs, and the reviewer's description is reasonable.

But in low resource settings, high dependency or intensive care beds are few, and it is very important that the patients at greatest risk are managed in these precious beds with experienced nursing staff. Ventilatory support and other intensive care aspects are not available on general wards, where most malaria is managed. The admitting physician therefore has to make rapid choices where and how the patient will be managed. This is an important determinant of outcome. This is the essence of triage, which is critical in busy, understaffed, under resourced hospitals, and it is informed by this simple and rapidly available test. The simple rapid test identifies those children who need the most attention. So, in short, in low resource settings, prognostic assessment is a very important determinant of hospital management and patient outcome.

3. A similar remark I have about the diagnostic value of pigmented PMNs. The authors state that a higher percentage of patients with hyperpigmented PMNs had high parasitic loads (583 out of 755) than of patients with lower pigmentation rates (1306 out of 2178). Similarly with my previous comment, if we were to use pigmented PMNs as diagnostic marker and treat those with high pigmentation rates differently from those without, then we may do more people harm ($1306/2933=45\%$) than good ($583/2933=20\%$).

This remark is about the value of diagnosis. We have shown in two previous papers (Watson et al. eLife 2021; Watson et al. Science Translation Medicine 2022) that about one third of children admitted to research centres (where we assume clinical evaluation is particularly good) with a diagnosis of severe malaria probably have another cause of their life-threatening illness (most likely to be bacterial sepsis as shown in Gilchrist et al. eLife 2022). This large mis-diagnosed subgroup has a high mortality. Although, on admission, sepsis and malaria may appear similar clinically, they are fundamentally different diseases. They have different complications requiring different management, so it is very important to distinguish the two. It is recommended that both antimalarials and antibiotics are given together, but in practice this advice is not followed. Finding $>5\%$ pigment containing neutrophils helps guide the clinician that the patient has “true” severe malaria and a high risk of dying from malaria complications. Finding low parasite counts and no pigment containing neutrophils encourages the clinician to look for sepsis and manage the patient accordingly. This is the value of rapid diagnosis.

4. Another way to look at the diagnostic value of the markers, is through the diagnostic accuracy paradigm (for which the authors used QUADAS-2, after all). With the numbers presented in the main text, we would have 583 true positives (high rate pigmented PMN and high parasitic load), 172 false positives (high rate+low parasites), 1306 false negatives (low rates + high parasitic load), 872 true negatives. That would lead to a sensitivity of $583/(583+1306)=31\%$ and a specificity of $872/(872+172)=84\%$. Which to me does not sound as a very promising diagnostic marker.

This is not the correct way to frame this question. All the patients have malaria parasitaemia – so all have “malaria” as currently defined, but some have incidental parasitaemia + another cause of severe illness as described above, whereas others have malaria as the primary cause of their life-threatening illness. Parasitaemia is much less specific than $>5\%$ pigment containing neutrophils as an indicator of “true severe malaria”. Thus, having $>5\%$ pigment containing neutrophils and a low parasite count is not a “false positive”.

Again, I am not saying that severe malaria is not important; I am also not saying that pigmentation of PMNs or monocytes is useless. But to state that they have strong prognostic and diagnostic value would be overstating the findings of this study, in my humble opinion.

Thank you. We have attenuated the discussion as requested.